# Quasi-periodic X-ray eruptions years after a nearby tidal disruption event

M. Nicholl[1✉], D. R. Pasham[2], A. Mummery[3], M. Guolo[4], K. Gendreau[5], G. C. Dewangan[6], E. C. Ferrara[5,7,8], R. Remillard[2], C. Bonnerot[9,10], J. Chakraborty[2], A. Hajela[11], V. S. Dhillon[12,13], A. F. Gillan[1], J. Greenwood[1], M. E. Huber[14], A. Janiuk[15], G. Salvesen[16], S. van Velzen[17], A. Aamer[1], K. D. Alexander[18], C. R. Angus[1], Z. Arzoumanian[5], K. Auchettl[19,20], E. Berger[21], T. de Boer[14], Y. Cendes[21,22], K. C. Chambers[14], T.-W. Chen[23], R. Chornock[24], M. D. Fulton[1], H. Gao[14], J. H. Gillanders[25], S. Gomez[26], B. P. Gompertz[9,10], A. C. Fabian[27], J. Herman[14], A. Ingram[28], E. Kara[2], T. Laskar[29,30], A. Lawrence[31], C.-C. Lin[14], T. B. Lowe[14], E. A. Magnier[14], R. Margutti[24], S. L. McGee[9,10], P. Minguez[14], T. Moore[1], E. Nathan[32], S. R. Oates[33], K. C. Patra[24], P. Ramsden[1,9,10], V. Ravi[32], E. J. Ridley[9,10], X. Sheng[1], S. J. Smartt[1,25], K. W. Smith[1], S. Srivastav[1,25], R. Stein[34], H. F. Stevance[1,25], S. G. D. Turner[35], R. J. Wainscoat[14], J. Weston[1], T. Wevers[26] & D. R. Young[1]

Quasi-periodic eruptions (QPEs) are luminous bursts of soft X-rays from the nuclei of galaxies, repeating on timescales of hours to weeks[1–5]. The mechanism behind these rare systems is uncertain, but most theories involve accretion disks around supermassive black holes (SMBHs) undergoing instabilities[6–8] or interacting with a stellar object in a close orbit[9–11]. It has been suggested that this disk could be created when the SMBH disrupts a passing star[8,11], implying that many QPEs should be preceded by observable tidal disruption events (TDEs). Two known QPE sources show long-term decays in quiescent luminosity consistent with TDEs[4,12] and two observed TDEs have exhibited X-ray flares consistent with individual eruptions[13,14]. TDEs and QPEs also occur preferentially in similar galaxies[15]. However, no confirmed repeating QPEs have been associated with a spectroscopically confirmed TDE or an optical TDE observed at peak brightness. Here we report the detection of nine X-ray QPEs with a mean recurrence time of approximately 48 h from AT2019qiz, a nearby and extensively studied optically selected TDE[16]. We detect and model the X-ray, ultraviolet (UV) and optical emission from the accretion disk and show that an orbiting body colliding with this disk provides a plausible explanation for the QPEs.

The TDE AT2019qiz was discovered by the Zwicky Transient Facility (ZTF) on 19 September 2019 UT (Universal Time), at RA 04 h 46 min 37.88 s and dec. −10° 13′ 34.90″ (J2000.0 epoch), in the nucleus of a barred spiral galaxy at redshift $z = 0.0151$ (luminosity distance of 65.6 Mpc). Its optical spectrum was typical of TDEs, with broad emission lines from hydrogen and ionized helium[16], and it is a particularly well-studied event owing to its proximity and early detection[16–18]. The UV and optical luminosity declined over a few months until reaching a steady, years-long plateau at about $10^{41}$ erg s$^{-1}$ (ref. 19), consistent with an exposed accretion disk[19,20]. Highly ionized iron lines appeared at this phase, indicating a gas-rich environment ionized by the TDE[21]. The mass of the central SMBH has been estimated as several $10^6\,M_\odot$ (in which $M_\odot$ is the solar mass) using various techniques (Extended Data Table 1).

We observed AT2019qiz on 9 and 10 December 2023 UT (approximately 1,500 days after its first optical detection) with the Chandra X-ray observatory and on 21 December 2023 UT with the Hubble Space Telescope (HST) as part of a joint programme to study TDE accretion disks. The Chandra data were obtained across three exposures of 15.4, 18.8 and 16.1 ks, shown in Fig. 1a. The average count rate in the Chandra broad band (0.5–7.0 keV) is more than an order of magnitude larger in the middle exposure than in the first and final exposures. The Chandra images show another X-ray source approximately 7 arcsec southeast (SE) of AT2019qiz, but the high spatial resolution of the Chandra images (about 0.5 arcsec) allows us to definitively associate the increase in count rate with AT2019qiz. The count rate increases and then decreases over the course of the middle exposure, whereas no other source in the field (Extended Data Fig. 1) shows evidence for variability. By analysing the spectra of these sources, we find that reported X-rays from the X-ray Telescope aboard the Neil Gehrels Swift Observatory (Swift/XRT) during the initial optical flare in 2019–2020 (ref. 16) are instead detections of the nearby SE source and we exclude these from any analysis in this work (Methods).

To further investigate the variability of AT2019qiz, we obtained high-cadence observations using the Neutron Star Interior Composition Explorer (NICER) from 29 February 2024 to 9 March 2024 UT, Swift/XRT on 12 March 2024 UT and AstroSat starting on 14 March 2024 UT. The soft X-ray (0.3–1.0 keV) light curves from NICER showed repeating sharp increases in count rate followed by a return to quiescence, with

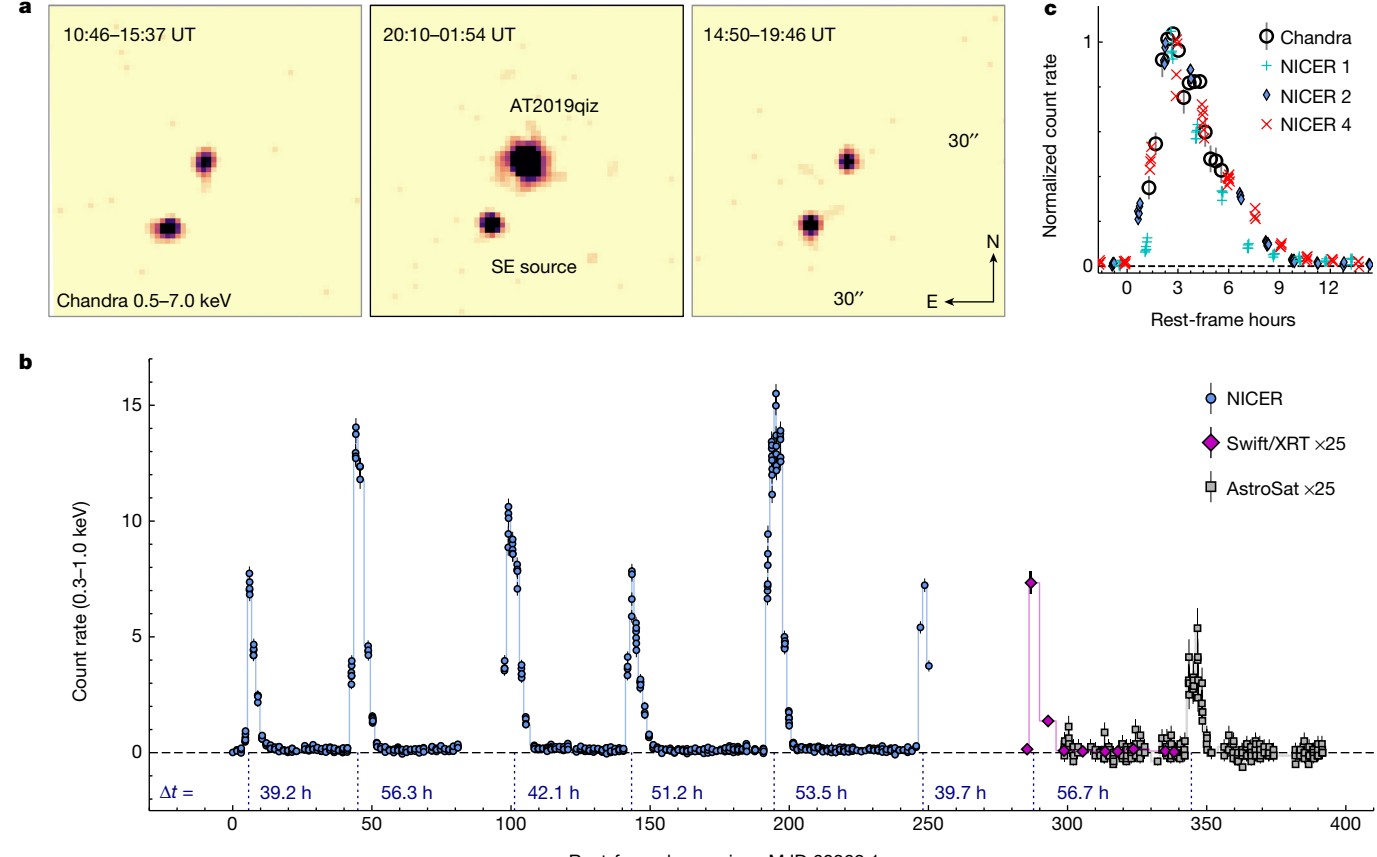

**Fig. 1 | Detection of QPEs from the nearby TDE AT2019qiz. a**, Chandra images obtained from exposures on 9 and 10 December 2023. Observation times are shown in UT. Each image shows a 30 × 30-arcsec region centred on AT2019qiz. Images have been smoothed with a 2-pixel Gaussian filter for clarity. The nearby source to the SE shows a consistent flux across the three exposures. **b**, Light curve showing eight eruptions detected by NICER, Swift/XRT and AstroSat from 29 February 2024 to 14 March 2024 (MJD 60369 to 60383).

Without stacking, the count rate between the eruptions is consistent with zero. Time delays between eruptions are labelled. The mean (standard deviation) recurrence time is 48.4 (7.2) h. **c**, Comparison of light-curve shapes between the Chandra eruption from December 2023 and NICER eruptions from March 2024. The fast rise and shallower decay remains consistent over several months. All error bars show $1\sigma$ uncertainties.

six consecutive peaks detected in just over 10 days. Two more peaks were detected over the next four days with Swift/XRT and AstroSat. The light curves are shown in Fig. 1b. The time between successive peaks ranges from 39 to 54 h in the rest frame, measured by fitting skewed Gaussian profiles (Extended Data Fig. 2). The mean recurrence time is 48.4 ± 0.3 h, with a standard deviation of 7.2 h. Typical durations are 8–10 h, with a consistent light-curve shape exhibiting a fast rise and slower decay (Fig. 1c).

The combination of soft X-ray sensitivity and cadence in the NICER data allows us to perform time-resolved spectral fitting (Fig. 2 and Extended Data Fig. 3). The nearby SE source detected by Chandra does not contribute substantially in the NICER bandpass (Methods). Single-temperature blackbody fits to the second NICER peak (chosen for good temporal coverage and low background; Methods) show an increasing temperature as the luminosity rises and a lower temperature for the same luminosity during the decay phase, owing to an increase in the blackbody radius. The expanding emitting region is approximately 1 solar radius (about $10^{11}$ cm). The bolometric luminosity at peak reaches $(1.8 \pm 0.1) \times 10^{43}$ erg s$^{-1}$, with a temperature of 109 ± 1 eV. In the quiescent phase, spectral information could only be retrieved by stacking the data from Swift/XRT. This can be well modelled as a colour-corrected disk model with maximum disk temperature $kT_\mathrm{p} \approx 67 \pm 10$ eV (Methods; Extended Data Fig. 4).

All of the above properties are consistent with the six known QPE sources repeating on timescales of hours to days (refs. 1–4) and the longer-duration Swift J0230+28 (refs. 5,22). This includes the luminosity and temperature, in both eruption and quiescence, and the lack of any detected optical/UV variability (Extended Data Fig. 5). The 'hysteresis loop' in the luminosity–temperature plane (Fig. 2c) is characteristic of QPE emission[12,23,24]. The recurrence time and eruption duration are towards the higher ends of their respective distributions (although well below Swift J0230+28), but their ratio of approximately 0.2 is consistent with the duty cycle of 0.24 ± 0.13 exhibited by other QPEs[5] (Fig. 3). Performing our own correlation analysis on duration versus recurrence time for the QPE population including AT2019qiz yields strong Bayesian evidence in favour of a correlation, with a mean duty cycle of $0.22^{+0.11}_{-0.04}$ (Methods). The roughly 15% variation in recurrence times in AT2019qiz is also similar to known QPEs. The variations in AT2019qiz seem irregular, but with a limited number of cycles, we cannot establish robustly at this point whether or not there is an underlying pattern of alternating long and short recurrence times, as seen in some of the other QPE sources[1,3].

We conclude that AT2019qiz is now exhibiting X-ray QPEs fully consistent with the known source population and with an average recurrence time $T_\mathrm{QPE} \approx 48$ h. Our result confirms theoretical predictions that at least some QPEs arise in accretion disks created by TDEs[8,11] (although we note that QPEs have also been discovered in galaxies with evidence for active nuclei[15]). It also increases confidence in the candidate QPEs following the TDEs AT2019vcb (ref. 14) and XMMSL1 J0249 (ref. 13) and the proposed X-ray TDE in the QPE source GSN 069 (ref. 12). We are

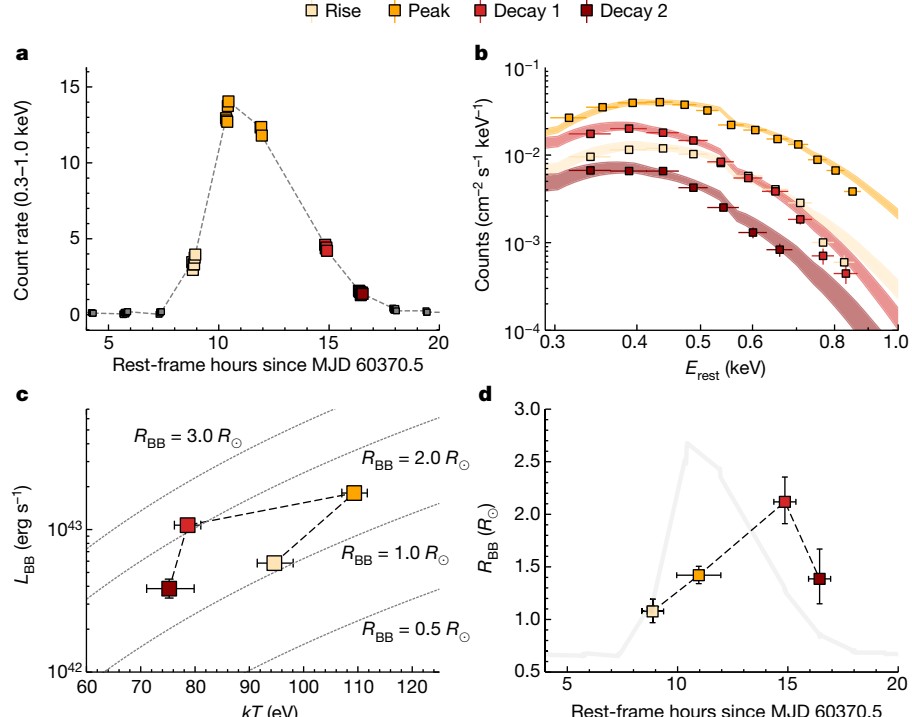

**Fig. 2 | NICER time-resolved spectroscopy of the second eruption in Fig. 1b. a**, Light curve of the eruption, with the rise, peak and decay phases indicated by the colour coding. **b**, Fits to the spectrum during each phase, using a single-temperature blackbody model (Methods). The shaded regions are 90% confidence intervals. **c**, Blackbody luminosity plotted against temperature for each fit. The eruption shows an anticlockwise 'hysteresis' cycle in this

parameter space. Error bars show the 90% confidence regions of the model posteriors. **d**, Blackbody radius against time, overlaid on the eruption light curve (grey). The blackbody radius increases during the eruption, with a maximum radius at the decay. We see tentative evidence in the final bin for contraction of the photosphere, which can be explained if the density and thus optical depth decrease as material expands.

unable to constrain when QPEs began in AT2019qiz, although NICER data in the two months around optical peak exhibit no QPEs. XRT data obtained on 13 January 2022 (about 840 days after disruption) over a duration of 25 h show the possible beginning of an eruption, but the duration of the observation is too short to confirm this (Methods; Extended Data Fig. 6).

Our HST imaging shows UV emission (effective wavelength 2,357 Å) coincident with the nucleus of the host galaxy. At this distance, the luminosity is $\nu L_\nu = 3.2 \times 10^{41}$ erg s⁻¹. This source is unresolved, indicating an angular size $\lesssim 0.08$ arcsec or 25 pc (Extended Data Fig. 7). The luminosity is consistent with a TDE accretion disk[20] but not with a nuclear star cluster (Methods). We also detect far-UV emission

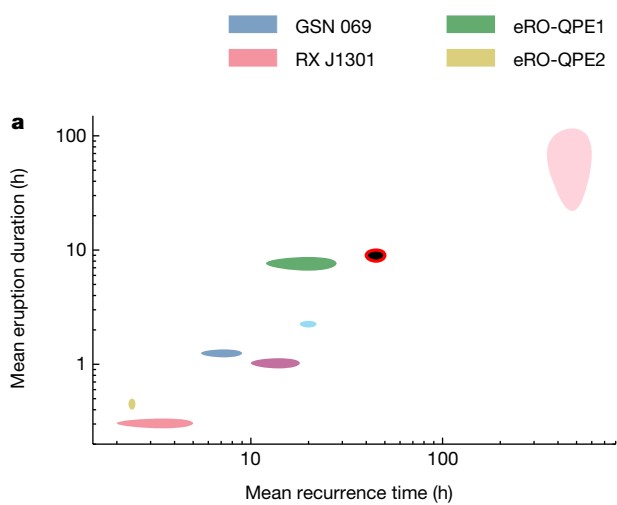

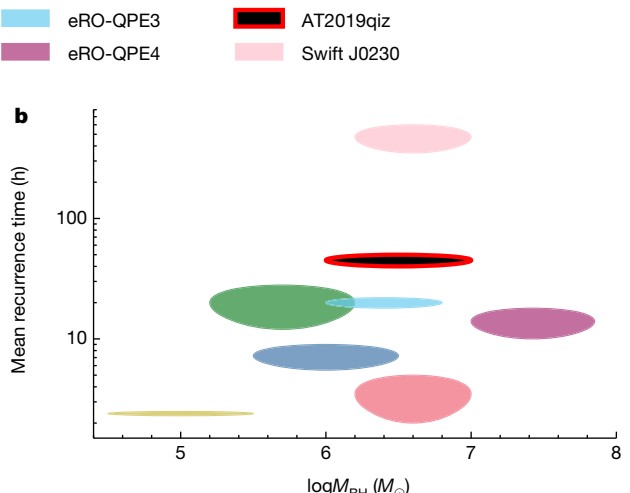

**Fig. 3 | Eruption properties in AT2019qiz compared with the other known QPE sources. a**, Mean eruption duration versus mean recurrence time. QPEs exhibit a clear correlation, with broader eruptions occurring for systems with longer recurrence times. The known QPE sources spend $24 \pm 13\%$ of their time in outburst[5]. AT2019qiz is consistent with this trend. **b**, Mean recurrence time versus reported SMBH mass from host galaxy scaling relations[5,16]. AT2019qiz is

completely typical of the known QPE population in terms of its SMBH mass and supports previous findings[5] that recurrence times in QPEs are not correlated with SMBH mass. The shaded regions represent the observed ranges of durations and recurrence times, whereas for the SMBH masses, they represent the $1\sigma$ uncertainty from scaling relations used to derive the masses[5].

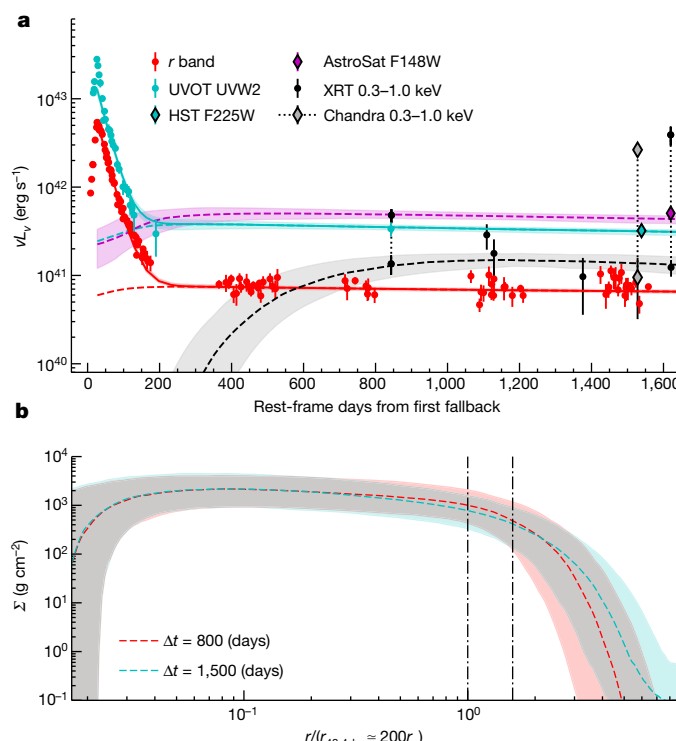

**Fig. 4 | Multiwavelength light curves with disk model fit. a**, X-ray, UV and optical data showing the TDE in 2019 (ref. 16) and the long-term disk emission. The dashed lines and shaded regions show the median and 90% confidence range of our accretion disk model fit[25]. QPEs (dotted lines) were excluded from the fit. A potential earlier QPE is also seen in the X-ray data at about 800 days (ref. 21). Our model is agnostic to the mechanism powering the initial UV/optical peak (Methods) but, by the time of the QPEs, all data are consistent with an exposed accretion disk. **b**, Radial surface density profiles of the best-fit model at 800 and 1,500 days after disruption (including 90% confidence range). The radius has been normalized to the circular orbit with period $T_{orb} = T_{QPE}$. The vertical lines indicate the orbital radii corresponding to periods of 1× and 2× $T_{QPE}$. Both orbits cross the disk plane, showing that star–disk interactions occurring either once or twice per orbit can explain the QPEs in AT2019qiz (ref. 11).

(1,480 Å) with AstroSat. We model the UV and quiescent X-ray light curves, alongside 3.5 years of optical measurements from the Panoramic Survey Telescope and Rapid Response System (Pan-STARRS) and ZTF, using a time-dependent relativistic thin disk[25] (Fig. 4; Methods). We find a SMBH mass $\log_{10}M_\bullet/M_\odot = 6.3^{+0.3}_{-0.2}$ and an initial disk mass $M_{disk}/M_\odot = 0.06^{+0.04}_{-0.03}$ (Extended Data Fig. 8).

The properties of the disk help to constrain the cause of the QPE emission. In models of disk-pressure instability, the variability amplitude and recurrence timescale depend on the SMBH mass and accretion rate. With the SMBH mass well constrained, the late-time disk luminosity is $(4 \pm 1)$% of the Eddington luminosity. At this Eddington ratio, radiation-pressure instability models can explain the amplitude of the eruptions but predict a recurrence time on the order of years[26]. A disk that is dominated by magnetic (rather than radiation) pressure is expected to be stable for this mass and Eddington ratio[8]. We therefore examine models that can explain QPE emission on hour to day timescales within a stable disk. These models involve another body (a star or compact object) already on a close, decaying orbit around the SMBH (an extreme-mass-ratio inspiral, or EMRI) that interacts with the spreading disk from the TDE once the disk is sufficiently radially extended.

The disk size is well constrained in our analysis by the UV and optical emission (Fig. 4) and is several times larger than an orbit with a

48.4-h period (radius approximately $200GM_\bullet/c^2$). Because any orbiting body with this period is expected to cross the disk, this provides a promising explanation for the observed QPEs. The same argument also applies to a 98.6-h orbit, required if interactions occur twice per orbit (Fig. 4). The luminosity in this model can be produced by the ejection of shocked disk material[11], shock breakout within the disk[27] or a temporarily enhanced accretion rate[28]. The compact emitting radius and its expansion during the eruptions may favour the first of these mechanisms. As the density of expanding ejecta decreases, we would expect the photosphere (the surface of the optically thick region) to eventually recede, consistent with our findings in Fig. 2d.

In the simplest case of an EMRI crossing the disk twice per elliptical orbit, recurrence times would exhibit an alternating long–short pattern, as seen in a subset of the known QPE sources[1,3]. In the EMRI model, more complex timing behaviour[2,23] can be caused by relativistic precession of the disk if its rotational axis is misaligned with that of the SMBH[10,29,30]. Notable precession over the course of a few cycles in AT2019qiz would require a dimensionless SMBH spin $a_\bullet \gtrsim 0.5$–0.7; however, such a large spin would tend to align the disk and damp precession in ≪1,000 days (Methods). Changing gas dynamics following star–disk collisions has recently been proposed as an alternative way to explain QPE timing variations[31]. Continuing high-cadence observations of AT2019qiz will be required to better constrain the nature of its timing variations and enable more detailed comparisons with QPE models.

The serendipitous discovery of QPEs in TDE AT2019qiz suggests that QPEs following TDEs may be common. We find that the long-term accretion disk properties in AT2019qiz are consistent with the star–disk interaction model for QPEs, indicating that the fraction of TDEs with QPEs can be used to constrain the rate of EMRIs, an important goal for future gravitational-wave detectors[32]. The latest observational estimates of the QPE rate[24] are about one-tenth of the TDE rate[33,34], consistent with recent theoretical predictions for the formation rate and lifetimes of EMRIs[35]. The QPEs in AT2019qiz show that long-term, high-cadence X-ray follow-up of optical TDEs will be a powerful tool for future QPE discovery, without the need for wide-field X-ray time-domain surveys, providing a path to measure the EMRI rate directly through electromagnetic observations.

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

[1]Astrophysics Research Centre, School of Mathematics and Physics, Queen's University Belfast, Belfast, UK. [2]Kavli Institute for Astrophysics and Space Research, Massachusetts Institute of Technology, Cambridge, MA, USA. [3]Clarendon Laboratory, Oxford Theoretical Physics, Oxford University, Oxford, UK. [4]Department of Physics and Astronomy, Johns Hopkins University, Baltimore, MD, USA. [5]Code 662, NASA Goddard Space Flight Center, Greenbelt, MD, USA. [6]Inter-University Centre for Astronomy and Astrophysics (IUCAA), Pune, India. [7]Department of Astronomy, University of Maryland, College Park, MD, USA. [8]Center for Research and Exploration in Space Science & Technology II (CRESST II), NASA Goddard Space Flight Center, Greenbelt, MD, USA. [9]School of Physics and Astronomy, University of Birmingham, Birmingham, UK. [10]Institute for Gravitational Wave Astronomy, University of Birmingham, Birmingham, UK. [11]DARK, Niels Bohr Institute, University of Copenhagen, Copenhagen, Denmark. [12]Department of Physics and Astronomy, University of Sheffield, Sheffield, UK. [13]Instituto de Astrofísica de Canarias, La Laguna, Tenerife, Spain. [14]Institute for Astronomy, University of Hawaii, Honolulu, HI, USA. [15]Center for Theoretical Physics, Polish Academy of Sciences, Warsaw, Poland. [16]Center for Theoretical Astrophysics, Los Alamos National Laboratory, Los Alamos, NM, USA. [17]Leiden Observatory, Leiden University, Leiden, The Netherlands. [18]Department of Astronomy and Steward Observatory, University of Arizona, Tucson, AZ, USA. [19]School of Physics, The University of Melbourne, Parkville, Victoria, Australia. [20]Department of Astronomy and Astrophysics, University of California, Santa Cruz, Santa Cruz, CA, USA. [21]Center for Astrophysics, Harvard & Smithsonian, Cambridge, MA, USA. [22]Department of Physics, University of Oregon, Eugene, OR, USA. [23]Graduate Institute of Astronomy, National Central University, Taoyuan, Taiwan. [24]Department of Astronomy, University of California, Berkeley, Berkeley, CA, USA. [25]Astrophysics, Department of Physics, University of Oxford, Oxford, UK. [26]Space Telescope Science Institute, Baltimore, MD, USA. [27]Institute of Astronomy, University of Cambridge, Cambridge, UK. [28]School of Mathematics, Statistics and Physics, Newcastle University, Newcastle upon Tyne, UK. [29]Department of Physics & Astronomy, The University of Utah, Salt Lake City, UT, USA. [30]Department of Astrophysics/IMAPP, Radboud University, Nijmegen, The Netherlands. [31]Royal Observatory, Institute for Astronomy, University of Edinburgh, Edinburgh, UK. [32]Cahill Center for Astronomy and Astrophysics, California Institute of Technology, Pasadena, CA, USA. [33]Department of Physics, Lancaster University, Lancaster, UK. [34]Division of Physics, Mathematics and Astronomy, California Institute of Technology, Pasadena, CA, USA. [35]Department of Applied Mathematics and Theoretical Physics, University of Cambridge, Cambridge, UK. [53]e-mail: matt.nicholl@qub.ac.uk

## Methods

### Observations and data analysis

**X-ray data. Chandra.** We downloaded processed Chandra images and event files and associated calibration data from the Chandra archive. We carried out analysis using CIAO version 4.16 (ref. 36) and CALDB version 4.11.0. We checked for pileup using the pileup_map task, finding a pileup fraction of about 1% only for the central 4 pixels of the middle exposure. Therefore, pileup has negligible impact on our analysis. Count rates were extracted using the srcflux task. We used a 2-arcsec (4-pixel) circular radius and the default point-spread function (PSF) model. The background was estimated using an annular region with inner and outer radii of 15 and 60 arcsec, respectively, centred on AT2019qiz. This excludes other point sources, including the SE source (see below). The CIAO srcflux task includes the Bayesian Gregory–Loredo algorithm[37] to determine the optimal number of bins for investigating a time-varying (or, more formally, periodic) signal. The algorithm provides an odds ratio for variability (2.5 for AT2019qiz) and a light curve with the number of bins that maximizes this odds ratio. None of the other five sources in Extended Data Fig. 1 show an odds ratio >1.

We extract the spectrum in both eruption and quiescence (see below) using the specextract task. The spectrum of the eruption is soft and can be reasonably fit with a blackbody of about 100 eV. We perform a more detailed spectral analysis of AT2019qiz using the later eruptions and quiescent-phase data from instruments with greater sensitivity to softer (0.3–0.7 keV) X-rays (see sections 'Swift/XRT and the quiescent spectrum of AT2019qiz' and 'NICER').

**The nature of the SE X-ray source.** The Chandra images show a nearby source approximately 7 arcsec to the SE (labelled 'SE source' in Fig. 1). It overlaps with the PSF of AT2019qiz in all instruments other than Chandra. We extracted individual X-ray (0.5–7.0-keV) spectra from all three Chandra obsIDs to characterize the SE source. We perform spectral analysis with the Bayesian X-ray Analysis (BXA) software version 4.0.7 (ref. 38), which connects the nested sampling algorithm Ultra-Nest[39] with the fitting environment XSPEC version 12.13.0c (ref. 40), in its Python version PyXspec. To improve the quality of the spectrum, we jointly fit all three Chandra obsIDs. The source can be fit with a simple power-law model with foreground absorption (tbabs × cflux(pow)) and is consistent with being constant over all three obsIDs. The neutral column density was fixed at the Milky Way value of $6.6 \times 10^{20}$ cm$^{-2}$. The 0.5–3.0-keV flux in the model is $2.1^{+1.6}_{-0.9} \times 10^{-14}$ erg s$^{-1}$ cm$^{-2}$ (90% posterior) and the photon index of the power law is $\Gamma = 1.8 \pm 0.5$ (90% posterior). The fit is shown in Extended Data Fig. 4a.

**Swift/XRT and the quiescent spectrum of AT2019qiz.** We obtained Target of Opportunity time to follow up AT2019qiz with Swift/XRT. Eleven observations were obtained from 12 March 2024 to 14 March 2024, with a typical exposure time of about 1,200 s per visit and cadence of 4.5 h. We clearly detect one eruption in the new data (Fig. 1). We also reanalysed all previous XRT data for this source obtained under previous programmes, using the online tools available through the UK Swift Science Data Centre[41,42].

Owing to the better sensitivity at soft energies compared with Chandra, we are able to model the underlying disk spectrum using the XRT observations during the quiescent phase. For this, we use a colour-corrected thermal disk model (tdediscspec)[43], to be consistent with the full spectral energy distribution (SED) fit (see section 'Disk modelling'). Given the larger PSF of XRT, we simultaneously model the AT2019qiz and the SE source contributions to the total spectrum. We use the model tbabs × (zashift(tdediscspec) + cflux(pow)), in which zashift(tdediscspec) is the contribution from AT2019qiz and cflux(pow) is the contribution from the SE source. The fit does not require a redshifted absorption component. We use PyXspec and BXA. For the disk parameters (that is, AT2019qiz), we assume flat priors; however, for the SE source, we use the posteriors from fitting its spatially resolved Chandra spectrum (see section 'The nature of the SE X-ray source') as

the priors. Extended Data Fig. 4b shows their individual contributions to the observed spectrum, confirming that AT2019qiz dominates at energies below $\simeq 1.0$ keV. The posteriors of the fit indicate a peak disk temperature $kT_p = 67 \pm 10$ eV (90% posterior), in agreement with the bulk TDE population[44].

**Archival data from Swift/XRT.** The X-ray spectrum of AT2019qiz observed by Swift/XRT in 2019–2020 was reported to be hard[16,21], suggesting a possible contribution from the SE source. To test this, we fit the combined spectrum (MJD 58714 to 59000) with the same power law plus disk model. We again use our power-law-fit posteriors for the SE source from Chandra as a prior in BXA and this time fix the temperature of the disk component while letting its flux vary freely. The early-time XRT spectrum is entirely consistent with the SE source, with no statistically significant contribution from the disk component (Extended Data Fig. 4c). This results in a $3\sigma$ upper limit on the flux (0.3–1.0 keV) from AT2019qiz at early times of $\leq 1.4 \times 10^{-14}$ erg s$^{-1}$ cm$^{-2}$, or a luminosity $\leq 7.2 \times 10^{39}$ erg s$^{-1}$.

By contrast, AT2019qiz is brighter and detected at high significance in data from 2022 onwards, with a spectrum dominated by the thermal component[21]. The luminosity of AT2019qiz measured during all quiescent phases with XRT and Chandra is roughly $10^{41}$ erg s$^{-1}$, more than an order of magnitude fainter than the eruptions. Extended Data Fig. 6 shows the observation from 2022 in bins of 5 ks. The final bin shows an increase in flux, but the temporal baseline is too short to confirm or rule out that this represents the onset of a QPE (see also Fig. 4). The spectral fit from ref. 21 is consistent with a blackbody with $kT_{BB} = 130 \pm 10$ eV, dominated by the final bin. We use the blackbody spectrum to calculate the luminosity in the final bin and exclude this bin from the disk model fit in Fig. 4a. We stack the remaining counts in a single bin and compute the quiescent luminosity using the fit from Extended Data Fig. 4.

**NICER.** NICER[45,46] observed AT2019qiz in two distinct campaigns, first at early times (around optical peak) from 25 September 2019 to 5 November 2019 and another at late times (about 1,600 days after optical peak) from 29 February 2024 to 9 March 2024.

The cleaned events lists were extracted using the standard NICER Data Analysis Software (HEASoft 6.33.2) tasks nicerl2 using the following filters: nicersaafilt=YES, saafilt=NO, trackfilt=YES, ang_dist=0.015, st_valid=YES, cor_range="*-*", min_fpm=38, underonly_range=0-80, overonly_range="0.0-1.0", overonly_expr="1.52*COR SAX**(-0.633)", elv=30 and br_earth=40. The whole dataset was acquired during orbit night-time and hence the daytime optical light leak (https://heasarc.gsfc.nasa.gov/docs/nicer/data_analysis/nicer_analysis_tips.html#lightleakincrease) does not apply to our data analysis. The latest NICER calibration release xti20240206 (6 February 2024) was used. Light curves in the 0.3–1.0-keV range were extracted using the nicerl3-lc task with a time bin size of 100 s and the SCORPEON background model.

The data obtained in the first campaign show no evidence for QPEs. Although the cadence is lower than that of the late-time data, it should be sufficient to detect QPEs occurring with the same frequency and duration as at late times, with a probability of detecting no QPEs of about 0.02 (using binomial statistics with a 20% duty cycle). We can therefore probably rule out QPEs within the first approximately two months after TDE fallback started (estimated to have occurred around 11 September 2019 (ref. 16)). However, we note that we would not expect QPEs during this phase in any model, as AT2019qiz was found to have an extended debris atmosphere[16], which remained optically thick to X-rays until much later[21].

During the second observing campaign, we clearly detect QPEs. The field of view of NICER is shown in Extended Data Fig. 1, overlaid on the Chandra image. All of the sources detected by Chandra have intensities (at energies lower than 1 keV) more than a factor of 10 below the measured peak of the QPE. Any contributions from these sources to the NICER spectra are further diminished by their offset angles from the centre of the field. We conclude that the NICER counts during

eruptions are completely dominated by AT2019qiz. The six consecutive eruptions detected by NICER were modelled using a skewed Gaussian fit to each peak (Extended Data Fig. 2). We measure rest-frame delay times of $39.3 \pm 0.3$, $56.3 \pm 0.3$, $42.1 \pm 0.3$, $51.2 \pm 0.2$ and $53.5 \pm 0.2$ h between successive eruptions.

Given the high count rate and good coverage, we extracted time-resolved X-ray spectra from the second NICER eruption (Fig. 1) in the 0.3–0.9-keV band. We created Good Time Intervals (GTIs) with nimaketime for four intervals representing the rise, peak and decay (two phases) of the eruption. We extracted these spectra using the nicerl3-spec task and produced SCORPEON background spectra in 'file mode' (bkgmodeltype=scorpeon bkgformat=file) for each of the four GTIs. We simultaneously fit the four spectra using PyXspec and BXA, assuming the model tbabs × zashift(bbody). We fixed the redshift to $z = 0.0151$ and included foreground absorption, with a neutral hydrogen column density fixed to $n_H = 6.6 \times 10^{20}$ cm$^{-2}$ (ref. 47). We initially included a redshifted absorber, but the model preferred zero contribution from this component, so we excluded it for simplicity. The full posteriors of the parameters are shown in Extended Data Fig. 3.

**AstroSat/SXT.** We observed AT2019qiz with AstroSat[48] for four days starting on 12 March 2024 UT using the Soft X-ray Telescope (SXT)[49] and the Ultra-Violet Imaging Telescope (UVIT)[50,51]. We used the level2 SXT data processed at the Payload Operation Centres using sxtpipeline v1.5. We merged the orbit-wise level2 data using SXTMerger.jl. We extracted the source in 200-s bins using a circular region of 12 arcmin. The broad PSF of the SXT does not leave any source-free regions for simultaneous background measurement. However, the background is low ($0.025 \pm 0.002$ counts s$^{-1}$) and steady. As the quiescent flux measured by Chandra is below the SXT detection limit, we take this count rate as our background estimate and subtract it from the light curve. SXT detected one eruption (MJD 60383.548).

**Optical/UV observations. HST.** We observed AT2019qiz using HST on 21 December 2023 UT (MJD 60299.55), obtaining one orbit with the Wide-Field Camera 3 (WFC3) UVIS channel in the F225W band. We downloaded the reduced, drizzled and charge-transfer-corrected image from the HST archive. We clearly detect a UV source coincident with the nucleus of the host galaxy. We verify that this source is consistent with a point source both by comparing the profile with other point sources in the image using the RadialProfile task in photutils and by confirming that the fraction of counts within apertures of 3 and 10 pixels are consistent with published encircled energy fractions in the UVIS documentation.

We perform aperture photometry using a 10-pixel (0.396-arcsec) circular aperture, measuring the galaxy background per square arcsecond using a circular annulus between 20 and 40 pixels and subtracting this from the source photometry. Although we cannot measure the galaxy light at the precise position of AT2019qiz, having no UV images free from TDE light, the estimated background within our aperture is <2% of the transient flux, so our results are not sensitive to this approximation. We correct to an infinite aperture using the encircled energy fraction of 85.8% recommended for F225W. The zero point is derived from the image header, including a chip-dependent flux correction. We measure a final magnitude of $20.63 \pm 0.03$ (AB).

Although the angular scale of about 25 pc is not small enough to rule out a nuclear star cluster (NSC), the UV source is an order of magnitude brighter than known NSCs[52]. Moreover, NSCs are generally red[53] and many magnitudes fainter than their host galaxies in bluer bands. The magnitude of the source we detect is comparable with the total UV magnitude of the galaxy[16]. An unresolved nuclear source was also detected in the QPE source GSN 069 (ref. 54).

**Ground-based photometry.** Numerous observations of this galaxy have been obtained by all-sky optical surveys both before and after the TDE. The optical emission was independently detected by ZTF[55,56], the Asteroid Terrestrial-impact Last Alert System (ATLAS)[57], Pan-STARRS[58] and the Gaia satellite[59].

Pan-STARRS reaches a typical limiting magnitude of about 22 in the broad $w$ filter (effective wavelength of 6,286 Å) in each 45-s exposure. All observations are processed and photometrically calibrated with the PS image-processing pipeline[60–62]. We downloaded and manually vetted all $w$-band observations of AT2019qiz since September 2019 and, in most cases, confirm a clean subtraction of the host galaxy light. We also retrieved ZTF forced photometry[63] in the $r$ band (with a similar effective wavelength of 6,417 Å). Owing to the shallower limiting magnitude of about 20.5, we stack the fluxes in 7-day bins. Both surveys clearly detect a continuing plateau, persisting for >1,000 days with a luminosity $vL_v \approx 7 \times 10^{40}$ erg s$^{-1}$. All Pan-STARRS and ZTF photometry was measured after subtraction of pre-TDE reference images using dedicated pipelines and hence include only light from AT2019qiz.

Although the optical light curves show scatter consistent with noise, they do not seem to exhibit the intense flaring behaviour seen in the X-rays. An order-of-magnitude flare in the optical would easily be detected even in the unbinned ZTF photometry. Assuming a duty cycle of 20%, and conservatively restricting to data since January 2022 (when we first see signs of day-timescale X-ray variability with XRT), the probability of never detecting an eruption simply because of gaps in cadence is $\lesssim 10^{-13}$.

To test for optical variability on shorter timescales, we conducted targeted observations with the 1.8-m Pan-STARRS2 telescope in Hawaii on 11 February 2024, with the IO:O instrument on the 2.0-m Liverpool Telescope[64] in La Palma on 15 February 2024 and with ULTRACAM[65] on the 3.5-m New Technology Telescope at the European Southern Observatory (La Silla) in Chile on 10 February 2024. Pan-STARRS images were obtained in the $w$ band (50 × 200-s exposures) and Liverpool Telescope in the $r$ band (32 × 120 s), whereas ULTRACAM observed simultaneously in the $u_s$, $g_s$ and $r_s$ bands[66] (384 × 20 s, with only 24 ms between exposures). All images were reduced through standard facility pipelines. For Pan-STARRS, this included subtraction of a pre-TDE reference image and forced photometry at the position of AT2019qiz. In the case of Liverpool Telescope and ULTRACAM, we performed photometry using psf[67], an open-source Python wrapper for photutils and other image-analysis routines. We excluded 17 ULTRACAM images affected by poor seeing. We attempted manual subtraction of the Pan-STARRS reference images using psf; however, we found that the extra noise introduced by the subtraction was larger than any detectable variability. As shown in Extended Data Fig. 5, there is no strong evidence for variability on timescales on the order of hours.

**Swift/UVOT.** UV observations were taken with Swift/UVOT in the uvm2 filter contemporaneously with the XRT observations. We used the uvotsource package to measure the UV photometry, using an aperture of 12″. We subtracted the host galaxy contribution by fitting archival photometry data with stellar population synthesis models using Prospector[68]. This standard procedure has been used to analyse previous UVOT observations of TDEs[56]. We apply Galactic extinction correction to all bands using a $E(B-V)$ value of 0.094 (ref. 69).

The UVOT photometry is shown in Extended Data Fig. 5. Although lacking the resolution of HST to separate the central point source from the host light, the mean measured magnitude of 20.1 is about 0.5 mag brighter than the host level estimated by SED modelling[16]. The individual measurements exhibit root-mean-square variation of 0.27 mag (Extended Data Fig. 5), possibly indicating variability that would further exclude a nuclear star cluster. The timing of the XRT QPE is marked, coinciding with a possible (but not statistically significant) dip in UV flux as seen in the QPE candidate XMMSL1 J0249 (ref. 13).

**AstroSat/UVIT.** We observed AT2019qiz with UVIT using the broad filter CaF2 (F148W)[50]. We processed the level1 data with the CCDLAB pipeline[70] and generated orbit-wise images, detecting a bright nuclear source. We performed aperture photometry using the UVITTools.jl package and the latest calibration[51], in a circular region of 20 pixels (8.2 arcsec). We also extracted background counts from a source-free area of the image. The background-corrected count rate in the merged

image corresponds to a flux density $f_\lambda = 3.16 \pm 0.97 \times 10^{-16}$ erg cm$^{-2}$ s$^{-1}$ Å$^{-1}$ or magnitude $m = 20.49 \pm 0.03$ (AB). We found no statistically significant far-ultraviolet variability between the orbit-wise images. We do not attempt to remove host galaxy flux for the UVIT data, as the field has not been covered by previous far-ultraviolet surveys. SED modelling would require a large extrapolation. Regardless, we expect that the galaxy flux should be negligible at these wavelengths[20].

## Analysis

**Assessing variability.** We perform two checks that the X-ray variability corresponds to QPEs rather than random variation. First we compare with physically motivated models of stochastic variability. Reference 71 demonstrated a mechanism to produce order-of-magnitude X-ray variability through Wien-tail amplification of accretion-disk perturbations. Their Fig. 3 shows the X-ray light curve of a model with a SMBH mass of $2 \times 10^6 M_\odot$, consistent with AT2019qiz. The light curves are of a visibly different character to our data, with random variability rather than flares of consistent duration and no obvious 'quiescent' level. We ran further simulations using their model and never found a light-curve segment resembling AT2019qiz.

We also take a model-agnostic approach and assume the null hypothesis that the times of the X-ray peaks are random. Drawing a list of $10^5$ delay times from a flat probability distribution between 0 and 60 h and examining every consecutive sequence of eight, we 'measure' the standard deviation in delay times to be ≤15% of the mean in only ≲0.1% of trials. This is not sensitive to where we place the upper and lower bounds of the distribution. Therefore, we can exclude random peak times at >3$\sigma$ confidence.

**QPE duration–recurrence time correlation.** The data in Fig. 3a show an apparent correlation between the mean duration and mean recurrence time of QPEs from a given source[5]. An equivalent statement is that QPEs seem to show a constant duty cycle across the population, with previous work indicating a duty cycle of 0.24 ± 0.13 (ref. 5). We reanalyse this correlation including AT2019qiz by performing Bayesian regression with a linear model $T_{\text{duration}} = \alpha T_{\text{recurrence}} + \beta$. We find $\alpha = 0.22^{+0.11}_{-0.04}$ (95% credible range), consistent with previous findings[5]. Comparing this model with the null hypothesis ($\alpha = 0$), we find a change in the Bayesian Information Criterion $\Delta$BIC $\approx 50$, indicating a strong preference for a positive linear correlation over the null hypothesis of no correlation.

**Disk modelling.** We use the time-dependent relativistic thin disk model developed in refs. 19,25. This computes the spectrum of an evolving accretion flow, produced at early times by the circularization of some fraction of the TDE stellar debris. To generate light curves, we follow the procedure of ref. 19 (their Fig. 2). The important input parameters are the mass and spin of the SMBH, the initial disk mass, the disk–observer inclination angle and the turbulent evolutionary timescale. Also, there are nuisance parameters relating to the initial surface density profile of the disk, which is generally unknown and has minimal effect on the late-time behaviour. As this initial condition is so poorly constrained, we simply consider an initial ring of material (as in ref. 25).

For each set of parameters $\{\Theta\}$, we compute the total (log-)likelihood

$$\mathcal{L}(\Theta) = -\sum_{\text{bands}, i} \sum_{\text{data}, j} \frac{(O_{i,j} - M_{i,j})^2}{E_{i,j}^2}, \tag{1}$$

in which $O_{i,j}$, $M_{i,j}$ and $E_{i,j}$ are the observed flux, model flux and flux uncertainty of the $j$th data point in the $i$th band, respectively. For the X-ray data, we compute the integrated 0.3–1.0-keV flux using the best-fit models to the quiescent Swift/XRT and Chandra data, whereas for optical/UV bands, we compute the flux at the effective frequency of the band. We correct all data for foreground extinction/absorption[47,69].

The early optical and UV observations do not examine direct emission from the accretion flow, because of either reprocessing[72] or shock emission from streams[73]. We add an early-time component to model out this decay[19], with functional form

$$L_{\text{early}} = L_0 \exp(-t/\tau_{\text{dec}}) \times \frac{B(\nu, T)}{B(\nu_0, T)}, \tag{2}$$

in which $B(\nu, T)$ is the Planck function and $\nu_0 = 6 \times 10^{14}$ Hz is a reference frequency. We fit the amplitude $L_0$, temperature $T$ and decay timescale $\tau_{\text{dec}}$, as well as the disk parameters. We only include data taken after the peak of the optical light curves.

The fit was performed using Markov chain Monte Carlo techniques, using the emcee formalism[74]. To speed up computations, analytic solutions of the relativistic disk equations[75] were used. The model satisfactorily reproduces all data. The model X-ray light curve shows a slow rise; however, this is completely unconstrained by data and is therefore very sensitive to the uncertain initial conditions of the simulation. After a few hundred days (by the time of the earliest X-ray data in Fig. 4), the disk has spread to large radii and is no longer sensitive to initial conditions. We present the posterior distributions of the physically relevant free parameters in Extended Data Fig. 8. The best-fitting SMBH mass is consistent with all other observational constraints.

We note that a dimensionless SMBH spin parameter $a_* > 0$ is favoured by the model (although see caveats below), with a peak in the posterior around $a_* \approx 0.4$–$0.5$. This constraint originates from the relative amplitudes of the optical/UV and X-ray luminosities, as highlighted in Extended Data Fig. 9. As the optical and UV light curves are well separated in frequency, the properties of the disk at scales $r \gtrsim 20 r_{\text{g}}$ are tightly constrained. The amplitude of the X-ray luminosity is controlled by the temperature of the inner disk, close to the innermost stable circular orbit. For a given large-scale structure, this radius is determined by $a_*$.

Our disk model parameterizes the colour correction factor $f_{\text{col}}$ in terms of the local disk temperature[76], but our posteriors do not marginalize over its unknown uncertainty. Recognizing that modest uncertainties in $f_{\text{col}}$ lead to substantial uncertainties in spin (for non-maximal black hole spins)[77], we do not claim a spin measurement here but simply note that a modest spin is consistent with our data. The spin estimates in this model also assume a planar disk that is aligned with the SMBH spin, which is not true in the case of a precessing disk (see next section).

Although the disk temperature profile (and therefore the location of the disk's outer edge) is tightly constrained from the multiband late-time observations, it is well known that disk temperature constraints only scrutinize the product $W^r_\phi \Sigma$, in which $W^r_\phi$ is the turbulent stress and $\Sigma$ is the surface mass density. As the functional form of the turbulent stress cannot be derived from first principles, and must be specified by hand, there is some uncertainty in the mid-disk density slope. Our choice of $W^r_\phi$ parameterization is optimized for computational speed[75] and is given by $W^r_\phi = w = $ constant. Rather than fit for $w$, we fit for the evolutionary timescale of the disk (which has a more obvious physical interpretation), given by $t_{\text{evol}} \equiv 2\sqrt{GM_* r_0^3}/9w$. We emphasize that this uncertainty has no effect on our constraints on the size of the disk.

With this choice of parameterization for the turbulent stress, the disk density profile (Fig. 4) can be approximated as $\Sigma \propto r^{-\zeta}$, with $\zeta = 1/2$, for $r = (2$–$600)GM_*/c^2$. The density slope is not very sensitive to modelling assumptions, with the (potentially) more physical radiation-pressure-dominated $\alpha$-disk model having $\zeta = 3/4$.

**Precession timescales.** If the SMBH is rotating, any orbit or disk that is misaligned with the spin axis will undergo Lense–Thirring precession. This is a possible cause of timing variations in QPEs[30]. Changes in QPE timing in AT2019qiz are seen over the course of ≲8 observed cycles,

which would require that the precession timescale $T_{\text{prec}}$ is approximately several $T_{\text{QPE}}$, in which $T_{\text{QPE}} \approx 48.4$ h is the QPE recurrence time.

The precession timescale can be calculated following[29]:

$$T_{\text{prec}} = \frac{8\pi G M_\bullet (1+2\zeta)}{c^3(5-2\zeta)} \frac{r_{\text{out}}^{5/2-\zeta} r_{\text{in}}^{1/2+\zeta}\left(1-(r_{\text{in}}/r_{\text{out}})^{5/2-\zeta}\right)}{a_\bullet\left(1-(r_{\text{in}}/r_{\text{out}})^{1/2+\zeta}\right)}, \qquad (3)$$

in which $r_{\text{in}}$ and $r_{\text{out}}$ are the inner and outer radii of the disk or orbit, respectively, in Schwarzschild units (see also ref. 78). We assume $\log(M_\bullet/M_\odot) = 6.3$ and investigate the plausible precession period for different values of $a_\bullet$.

The nodal precession timescale for an orbiting body can be estimated by calculating $T_{\text{prec}}$ at the orbital radius (setting $R_{\text{in}} \approx R_{\text{out}} \approx R_{\text{orb}}$). For $a_\bullet = 0.1$–$0.9$, this gives $T_{\text{prec,orbit}} \approx (10^3$–$10^4) \times T_{\text{QPE}}$, independent of $\zeta$. Therefore, in the EMRI model, nodal precession is too slow to account for changes in QPE timing over several orbits.

The precession timescale of the disk can be calculated by assuming that it behaves as a rigid body with $r_{\text{in}} = 2GM_\bullet/c^2$, $r_{\text{out}} = 600GM_\bullet/c^2$ and a density slope $\zeta = 1/2$ from our disk model. We use the above equation to find $T_{\text{prec,disk}} \approx (70$–$200) \times T_{\text{QPE}}$ (for the same range of spins). With a steeper density profile having $\zeta = 1$, this would reduce to $T_{\text{prec,disk}} \approx (8$–$70) \times T_{\text{QPE}}$ (because more mass closer to the SMBH enables stronger precession). Therefore, precession can explain detectable changes in QPE timing over the course of several orbits only in the case of a rapidly spinning SMBH ($a_\bullet \gtrsim 0.5$–$0.7$) and a steep disk density profile.

With these constraints, attributing the timing residuals primarily to disk precession becomes challenging. The larger the SMBH spin magnitude, the faster an initially inclined disk will come into alignment with the black hole spin axis, damping precession on a timescale $\lesssim 100$ days for $a_\bullet > 0.6$ and $M_\bullet \approx 10^6 M_\odot$ (ref. 79). To maintain precession for more than 1,000 days requires a spin $a_\bullet \lesssim 0.2$, in which case the precession is not fast enough to fully explain the timing variations in our data.

We also note that the disk inner radius used in our precession calculation was derived from a planar disk model. In a tilted disk around a spinning SMBH, the radius of the innermost stable circular orbit will differ from the equatorial case. Understanding the effect of disk precession in AT2019qiz will probably require both continued monitoring to better understand the QPE timing structure and a self-consistent model of an evolving and precessing disk that can explain both the multiwavelength light curve and the timing residuals.

**Constraints on QPE models.** Many models have been proposed to explain QPEs. Disk tearing owing to Lense–Thirring precession has been suggested[80]. This effect has plausibly been detected in the TDE AT2020ocn (ref. 81). However, its X-ray light curve did not resemble that of AT2019qiz or those of other QPEs. As discussed above, it is also unclear whether strong precession will persist until such late times. The X-ray variability in AT2020ocn occurred only in the first months following the TDE.

Gravitational lensing of an accretion disk by a second SMBH in a tight binary could cause periodic X-ray peaks for the right inclination[82]. However, in the case of AT2019qiz, no signs of gravitational self-lensing were detected during the initial TDE. In this model, a QPE magnification by a factor $\gtrsim 10$ requires an extremely edge-on view of the disk, which leads to a shorter duration of the QPE flares. This was already problematic for previous QPEs[82] and is more so for the longer-duration flares in AT2019qiz. Moreover, finding a TDE around a close SMBH binary within a very narrow range of viewing angles ($\gtrsim 89.5°$) is very unlikely in the small sample of known TDEs, so a strong TDE–QPE connection is not expected in this model.

Limit-cycle instabilities are an appealing way to explain recurrent variability[7,83]. The recurrence timescale for disk-pressure instabilities depends on whether the disk is dominated by radiation pressure or magnetic fields[8], as well as the accretion rate. Our disk model, which

is well constrained by the multiwavelength data, gives an Eddington ratio $\dot{M}/M_{\text{Edd}} \approx L/L_{\text{Edd}} = 0.04 \pm 0.01$. Reference 26 gives formulae to interpolate the recurrence time for radiation-pressure instabilities, for a given amplitude relative to quiescence. We assume a peak-to-quiescence luminosity ratio of 60, although our analysis is not sensitive to this. Using the prescription for either the intermediate-mass black holes (their equation 33) or SMBHs (their equation 34), we find a recurrence time of about 5,000 days.

In the magnetic case, we use equation 14 from ref. 8. Matching the observed recurrence time requires a dimensionless magnetic-pressure scaling parameter $p_0 \approx 10$. However, at this Eddington ratio, the disk should be stable[8] if $p_0 \gtrsim 1$. This leaves no self-consistent solution in which magnetic-pressure instabilities cause the QPEs in AT2019qiz. The possibility of a long–short cycle in recurrence time, and the asymmetric profile of the eruptions[3], also disfavour pressure instabilities. We also note that, in disk-instability models, the recurrence time of the instability correlates with SMBH mass. For the known QPEs, there is no apparent correlation in recurrence time with mass (Fig. 3).

The final class of models to explain QPEs involves an orbiting body (EMRI) either transferring mass to an accretion disk or colliding with it repeatedly[9–11,27,28,30,35,84–86]. Note that this is very unlikely to be the same star that was disrupted during the TDE: if a bound remnant survived the disruption, it is expected to be on a highly eccentric orbit with a much longer period than the QPEs[11]. The fundamental requirement for star–disk collisions to explain QPEs is that the disk is wider than the orbit of the EMRI. The size of the disk in AT2019qiz is well constrained by our analysis and the posteriors of our fit fully satisfy this requirement, at least in the case of a circular disk.

For an orbit with the QPE period to avoid intersecting the disk would require a disk ellipticity $e > 0.7$ (assuming that the semimajor axis of the disk is fixed) and an appropriately chosen orbital inclination. Although some TDE spectra support a highly elliptical disk in the tens of days after disruption[87], most can be explained with an approximately circular disk[88–90]. Simulations of TDE accretion disks show a high ellipticity in the days after disruption[91], but shocks are expected to circularize the disk over the course of a few debris orbital periods[92] (days to weeks), whereas we observe QPEs on a timescale of years after AT2019qiz. An initially highly eccentric disk becomes only mildly elliptical ($e \approx 0.6$) on a timescale of several days (refs. 93–100). Once notable fallback has stopped (before the plateau phase), no more eccentricity will be excited in the disk, whereas turbulence will act to further circularize it, so we expect that the disk in AT2019qiz will be circular to a good approximation.

The case of an EMRI interacting with a TDE disk was specifically predicted by refs. 11,30. The formation rate of EMRIs by the Hills mechanism is about $10^{-5}$ year$^{-1}$ galaxy$^{-1}$, about one-tenth of the TDE rate. Because the time for inspiral resulting from gravitational-wave emission (approximately $10^6$ years) is longer than the time between TDEs (approximately $10^4$ years), theory predicts that $\gtrsim 1$ in 10 TDEs could host an EMRI capable of producing QPEs[11,35]. This is consistent with recent observational constraints on the QPE rate[24].

## Data availability

All NICER, Chandra and Swift data presented here are public and can be found in the NASA archives at https://heasarc.gsfc.nasa.gov/cgi-bin/W3Browse/w3browse.pl. HST data are public through the MAST archive at https://archive.stsci.edu/missions-and-data/hst. The reduced light-curve data from Figs. 1 and 4 are available on GitHub at https://github.com/mnicholl/AT2019qiz.

## Code availability

Data reduction and X-ray spectral fitting were performed using standard, publicly available codes (Methods). Code used for the relativistic

disk model is described by refs. 19,25. Author A.M. is working towards releasing a user-friendly version of this code publicly through GitHub; the current version will be shared on request.

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

**Acknowledgements** We thank the Swift, AstroSat and NICER teams for scheduling our DDT requests. We thank the participants of the Kavli Institute for Theoretical Physics 'TDE24' meeting and C. Done for helpful discussions. M.N., A.A., C.R.A. and X.S. are supported by the European Research Council (ERC) under the European Union's Horizon 2020 research and innovation programme (grant agreement no. 948381) and by UK Space Agency grant no. ST/Y000692/1. D.R.P. was supported by NASA grant 80NSSC19K1287. This work was supported by a Leverhulme Trust International Professorship grant (number LIP-202-014). E.C.F. is supported by NASA under award number 80GSFC21M0002. A.H. is supported by Carlsberg Foundation Fellowship Programme 2015. V.S.D. and ULTRACAM are financed by the UK Science and Technology Facilities Council (STFC, grant ST/Z000033/1). A.J. is supported by grant no. 2023/50/A/ST9/00527 from the Polish National Science Centre. E.J.R. and P.R. are supported by STFC studentships. K.D.A. acknowledges support from the National Science Foundation through award AST-2307668. K.A. is supported by the Australian Research Council Discovery Early Career Researcher Award (DECRA) through project number DE230101069. T.-W.C. acknowledges the Yushan Young Fellow Program by the Ministry of Education, Taiwan for the financial support. R.C. benefited from interactions with Theory Network participants that were supported by the Gordon and Betty Moore Foundation through grant GBMF5076. K.C.P. is financed in part by generous support from S. Nagaraj, L. Noll and S. Otellini. E.N. acknowledges support from NASA theory grant 80NSSC20K0540. A.I. acknowledges support from the Royal Society. S.G.D.T. acknowledges support under STFC grant ST/X001113/1. A.F.G. acknowledges support from the Department for the Economy (DfE) Northern Ireland postgraduate studentship scheme. This research was supported in part by grant NSF PHY-2309135 to the Kavli Institute for Theoretical Physics. This research has made use of data obtained from the Chandra Data Archive and the Chandra Source Catalog, and software provided by the Chandra X-ray Center (CXC) in the application packages CIAO and Sherpa. The AstroSat mission is operated by the Indian Space Research Organisation (ISRO) and the data are archived at the Indian Space Science Data Centre (ISSDC). The SXT data-processing software is provided by the Tata Institute of Fundamental Research (TIFR), Mumbai, India. The UVIT data were checked and verified by the UVIT POC at IIA, Bangalore, India. We acknowledge the use of public data from the Swift data archive. The Pan-STARRS telescopes are supported by NASA grants NNX12AR65G and NNX14AM74G. ZTF is supported by the National Science Foundation under grant nos. AST-1440341 and AST-2034437 and a collaboration including current partners Caltech, IPAC, the Oskar Klein Center at Stockholm

University, the University of Maryland, University of California, Berkeley, the University of Wisconsin at Milwaukee, University of Warwick, Ruhr University, Cornell University, Northwestern University and Drexel University. Operations are conducted by COO, IPAC and UW. The Liverpool Telescope is operated on the island of La Palma by Liverpool John Moores University in the Spanish Observatorio del Roque de los Muchachos of the Instituto de Astrofisica de Canarias with financial support from the UK STFC.

**Author contributions** M.N. was PI of the Chandra, Hubble Space Telescope, Swift and Liverpool Telescope programmes, performed data analysis and led the writing and overall project. D.R.P. prompted the NICER and AstroSat follow-up, performed X-ray data reduction and analysis and wrote parts of the manuscript. A.M. performed the accretion disk modelling and wrote parts of the manuscript. M.G. performed X-ray data reduction and spectral analysis, the comparison with other QPE sources and wrote parts of the manuscript. K.G. and E.C.F. coordinated the NICER observations. K.G. is PI of NICER. G.C.D. coordinated AstroSat observations and reduced the data. R.R. performed NICER data reduction. C.B. and J.C. contributed to the precession analysis. A.H. performed Chandra data reduction. V.S.D. organized and reduced the ULTRACAM observations. A.F.G. analysed the HST PSF. J.G. analysed the Pan-STARRS plateau. M.E.H. obtained the Pan-STARRS observations. A.J. contributed to pressure instability models. G.S. analysed the SMBH spin systematics. S.v.V. contributed the ZTF forced photometry. A.A., K.D.A., K.A., E.B., Y.C., R.C., S.G., B.P.G., T.L., A.L., R.M., S.L.M., S.R.O., E.J.R. and X.S. contributed to the Chandra+HST programme. Z.A., A.C.F., E.C.F., K.G., E.K. and R.R. are members of the NICER team. Z.A. and K.G. carried out the NICER observations. A.A., C.R.A., T.-W.C., M.D.F., J.H.G., T.M., P.R., X.S., S.J.S., K.W.S., S.S., H.F.S., J.W. and D.R.Y. are members of the Pan-STARRS Transients Team. T.d.B., K.C.C., H.G., J.H., C.-C.L., T.B.L., E.A.M., P.M., S.J.S., K.W.S., R.J.W. and D.R.Y. contributed to the operation of Pan-STARRS. A.I., E.N. and S.G.D.T. provided theoretical expertise. R.C., R.M., K.C.P., V.R. and R.S. are members of the ZTF TDE group. T.W. provided HST expertise. All authors provided feedback on the manuscript.

**Competing interests** The authors declare no competing interests.

**Additional information**
**Correspondence and requests for materials** should be addressed to M. Nicholl.

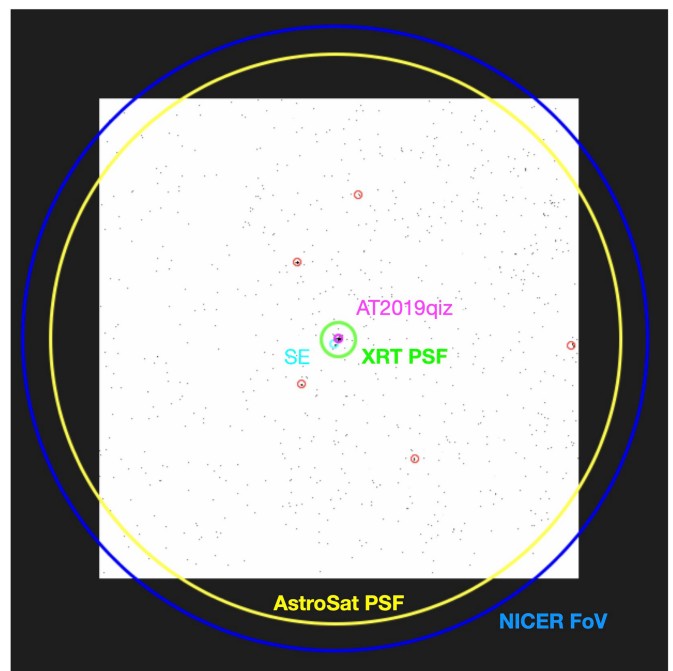

**Extended Data Fig. 1 | Chandra image during eruption.** The image is
8.5 × 8.5 arcmin, with north up and east to the left. Five sources are detected
within a few arcminutes of AT2019qiz. Only AT2019qiz shows statistical
evidence of variability in the Chandra data (Methods). The PSFs (half-encircled
energy width) of Swift/XRT and AstroSat are marked, as is the NICER field of
view. None of the sources exhibit a count rate (0.3–1.0 keV) above about 10%
of the count rate from AT2019qiz during eruption. Figure 1a shows a zoom-in
of the central region.

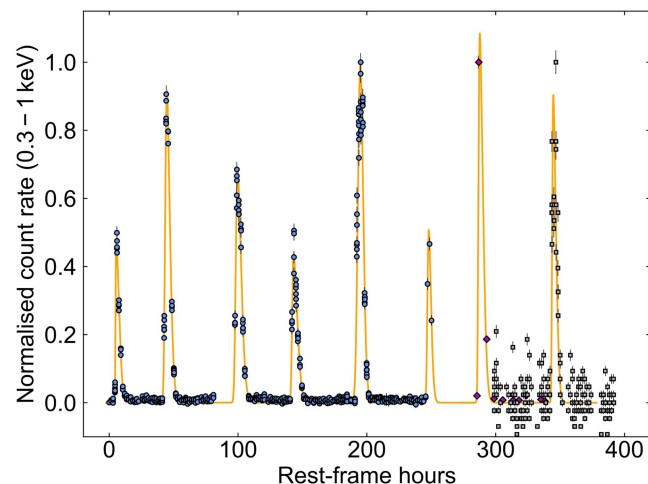

**Extended Data Fig. 2 | Estimates of the peak times of each eruption.** Each peak has been fit separately with a skewed Gaussian function using SciPy. This takes four parameters: the mean $\mu$ of the unskewed Gaussian, the standard deviation $\sigma$, the skewness $a$ and an arbitrary normalization. We take the maximum of the function as the time of each peak. The uncertainty in timing is given by the variance in $\mu$. The error bars show the $1\sigma$ uncertainty in count rate.

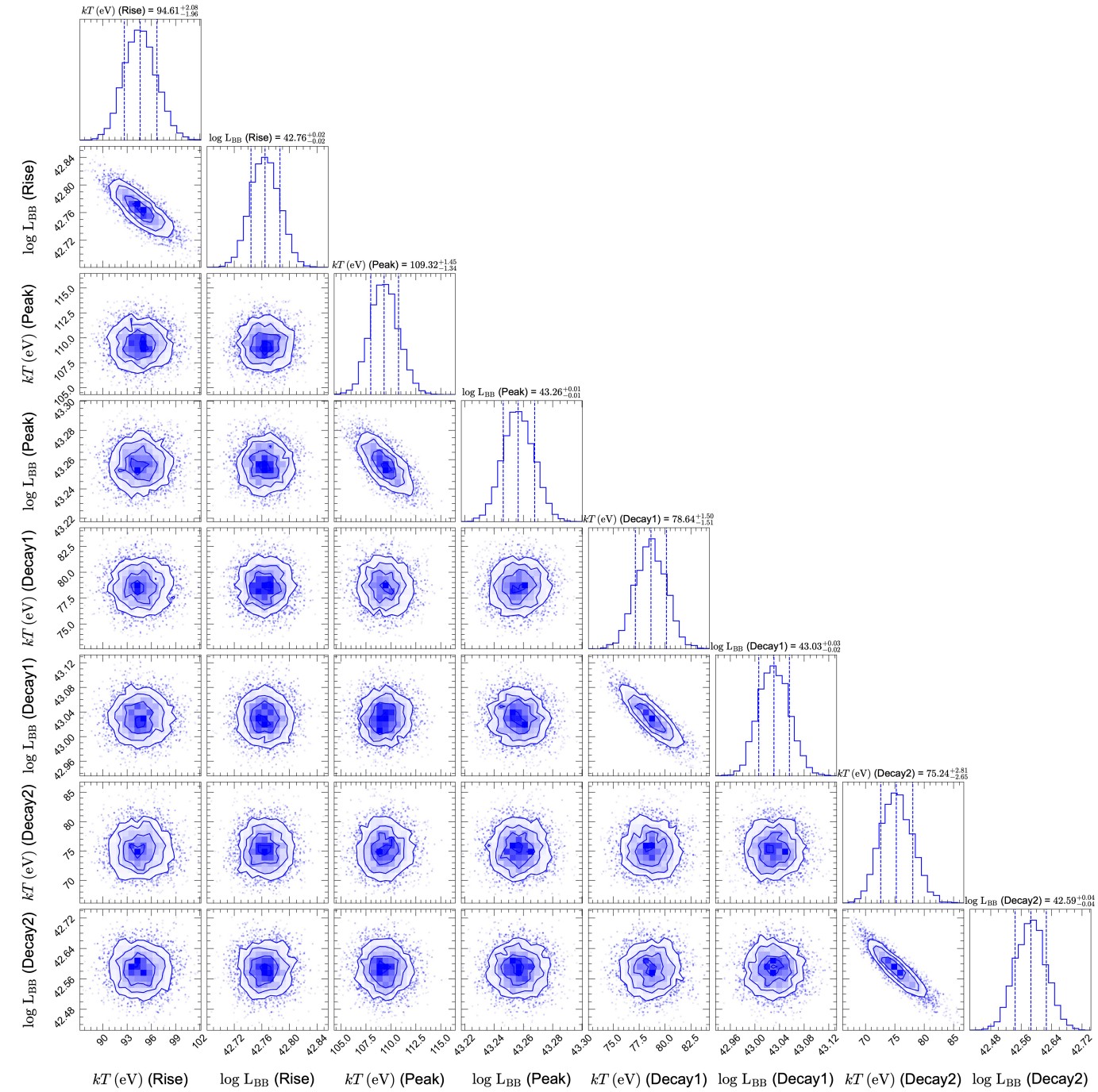

**Extended Data Fig. 3 | Physical parameters from the second eruption detected with NICER.** Corner plot showing posterior distributions of all free parameters from the time-resolved spectral modelling of the second NICER eruption (Fig. 2).

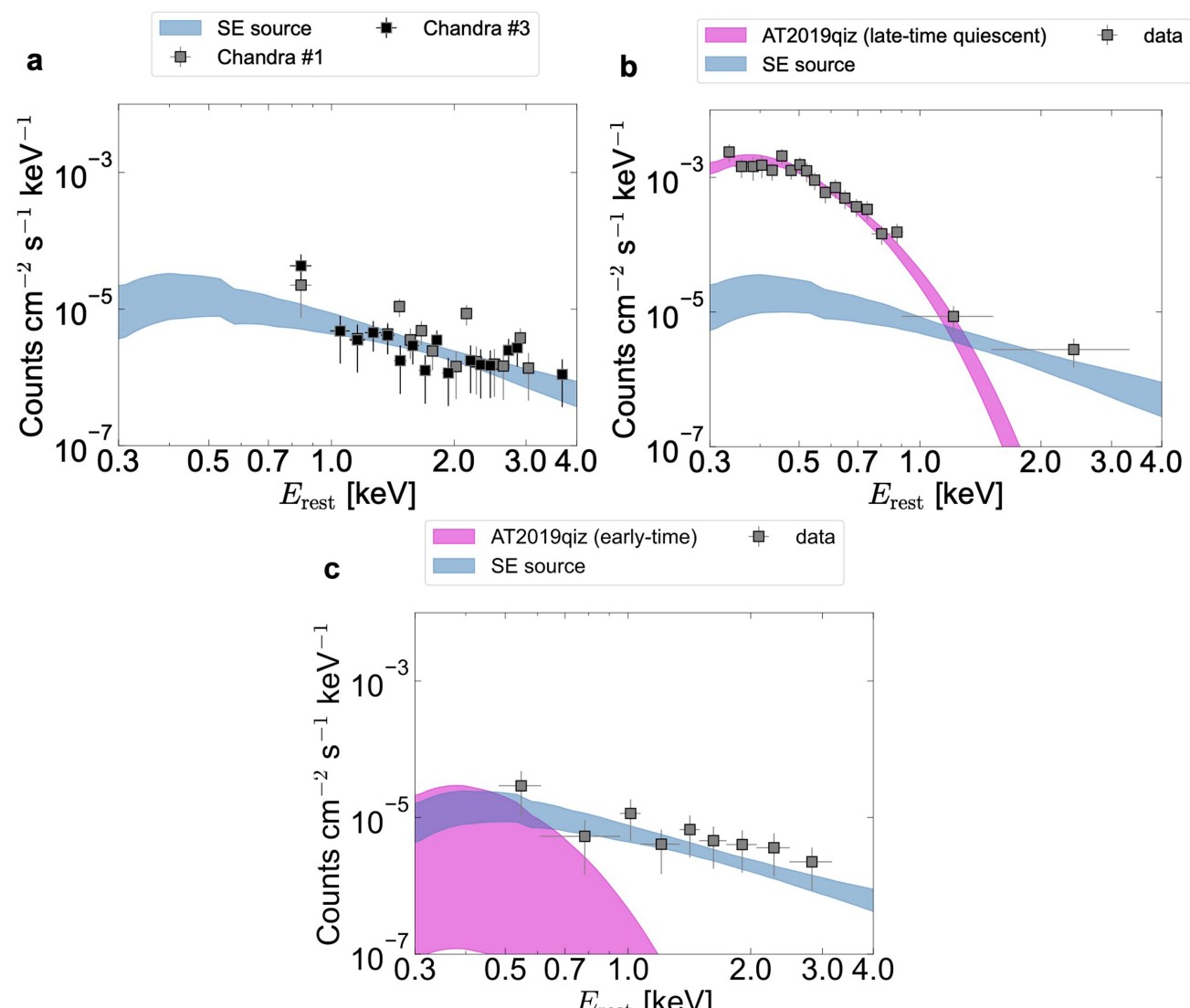

**Extended Data Fig. 4 | Fits to the quiescent spectrum of AT2019qiz and the nearby SE source.** Shaded regions show 90% confidence intervals. **a**, Fit to the SE source from Chandra (first and third epochs). The data are best fit with a power law with $\Gamma = 1.8 \pm 0.5$. **b**, Fit to the quiescent spectrum from Swift/XRT. This includes flux from both sources. We fit with a power law plus a thermal disk model including colour correction (tdediscspec), using the posteriors from the SE source as the priors on the power-law component. The SE source clearly dominates the count rate above $\simeq$1 keV. Below this, the spectrum is well fit by the thermal disk with peak temperature $kT_{\mathrm{p}} = 67 \pm 10$ eV, similar to other QPE sources during their quiescent phases[1,3] and similar to X-ray-detected TDEs[44]. The SE source contribution is shown in blue. **c**, Fit to the X-ray spectrum during the initial phase of the TDE optical component (MJD 58714 to 59000) using the temperature and power-law slope from panels **a** and **b**. The spectrum is consistent with emission from the SE source, with no statistically significant contribution from AT2019qiz.

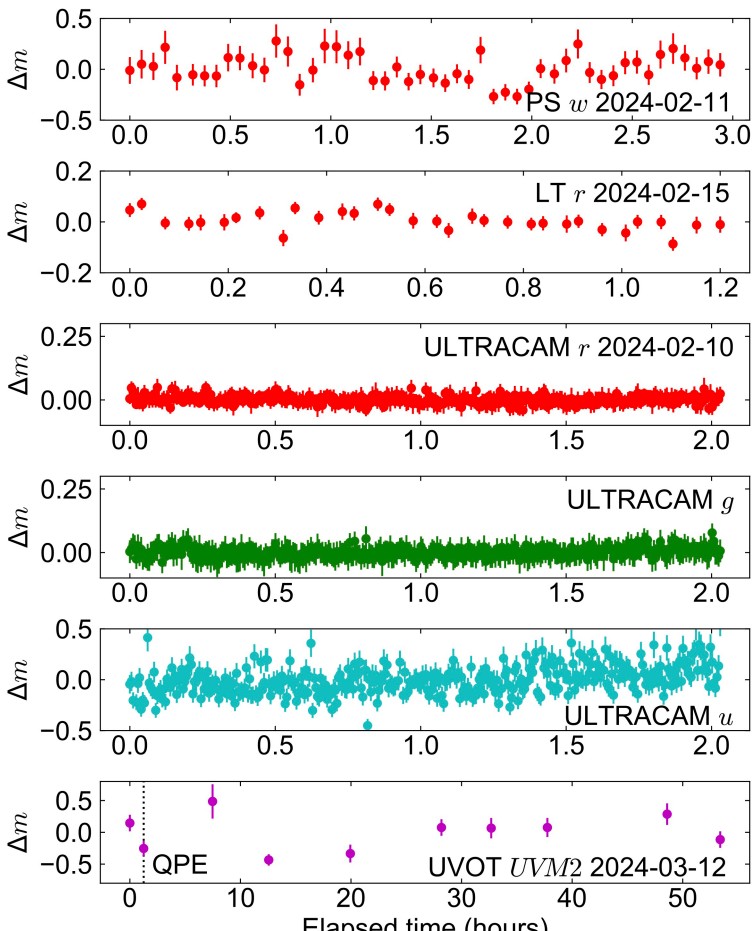

**Extended Data Fig. 5 | High-cadence optical observations and UV photometry.** Pan-STARRS data are measured on difference images using the Pan-STARRS reference image for subtraction, whereas for the Liverpool Telescope, ULTRACAM and UVOT, we measure aperture photometry on the unsubtracted images. We subtract the mean magnitude in each case to emphasize the (lack of) strong variability on hour-long timescales. However, the UV shows possible variability at the level of several times 0.1 mag, with a possible dip at the time of the QPE[13]. Note that the time axis is different on each sub-plot and the dates on which each dataset was obtained are provided on the individual panels. The error bars show the 1$\sigma$ uncertainty in magnitude.

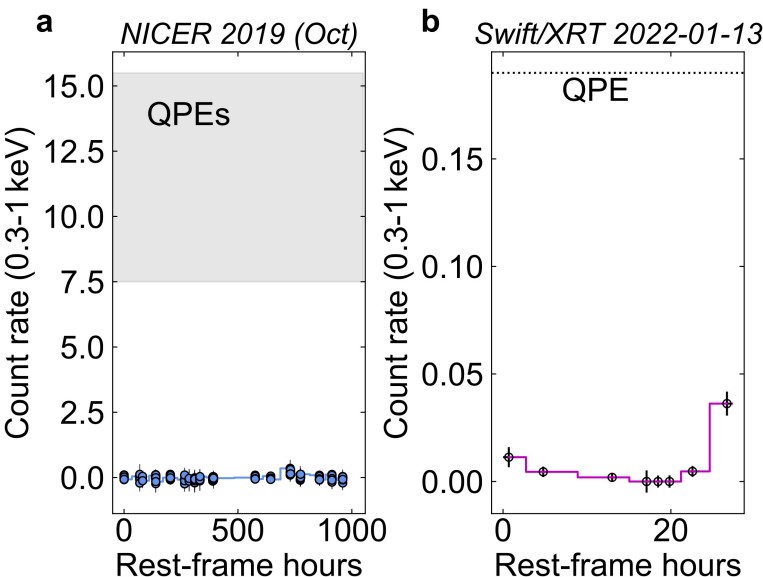

**Extended Data Fig. 6 | High-cadence X-ray observations at earlier times.**
**a**, NICER data obtained between 25 September 2019 and 5 November 2019, close to the time of the optical TDE peak. No variability is detected, with the shaded region showing the range of QPE peaks at late times. Comparing with the observed QPEs at late times suggests that we would have most probably detected about two QPEs if they were active. **b**, Swift/XRT data[21] obtained on 13 January 2022, binned in 5-ks fixed bins. The dotted line shows the QPE detected later with XRT. Variability is now observed on approximately hour timescales, but the baseline is insufficient to determine whether this is QPE-like in nature. The error bars show the $1\sigma$ uncertainty in count rate.

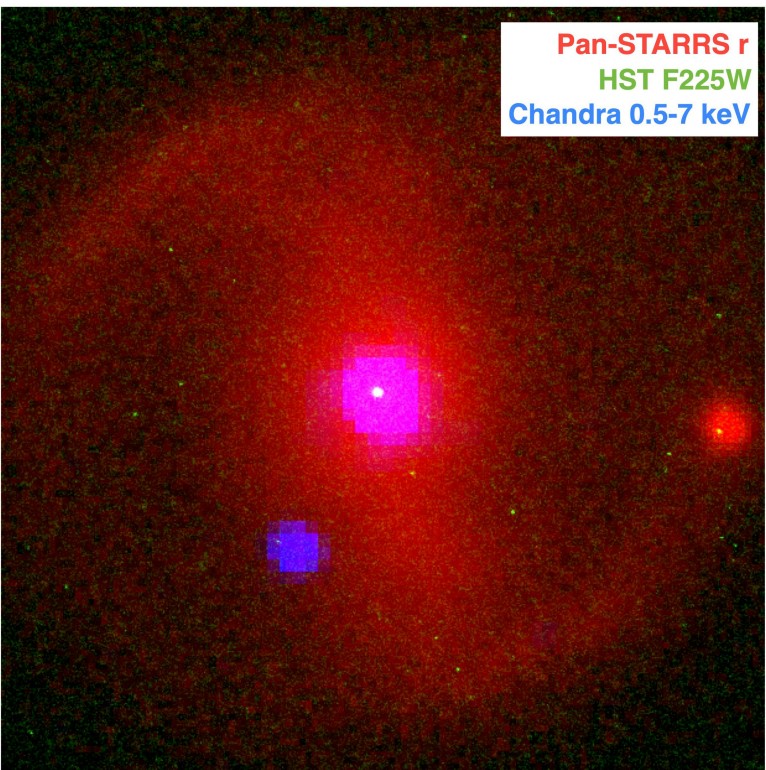

**Extended Data Fig. 7 | Optical, UV and X-ray images.** 30 × 30-arcsec false-colour image, centred at the position of AT2019qiz. The red channel shows the archival Pan-STARRS stacked image of the field in the *r* band. The blue channel shows the Chandra image during the QPE (which appears magenta overlaid on the red Pan-STARRS image), smoothed with a 2-pixel Gaussian filter. The green channel shows the HST image, demonstrating the point nature of the UV emission (visible as a white dot at the centre of the image) and its association with the host nucleus.

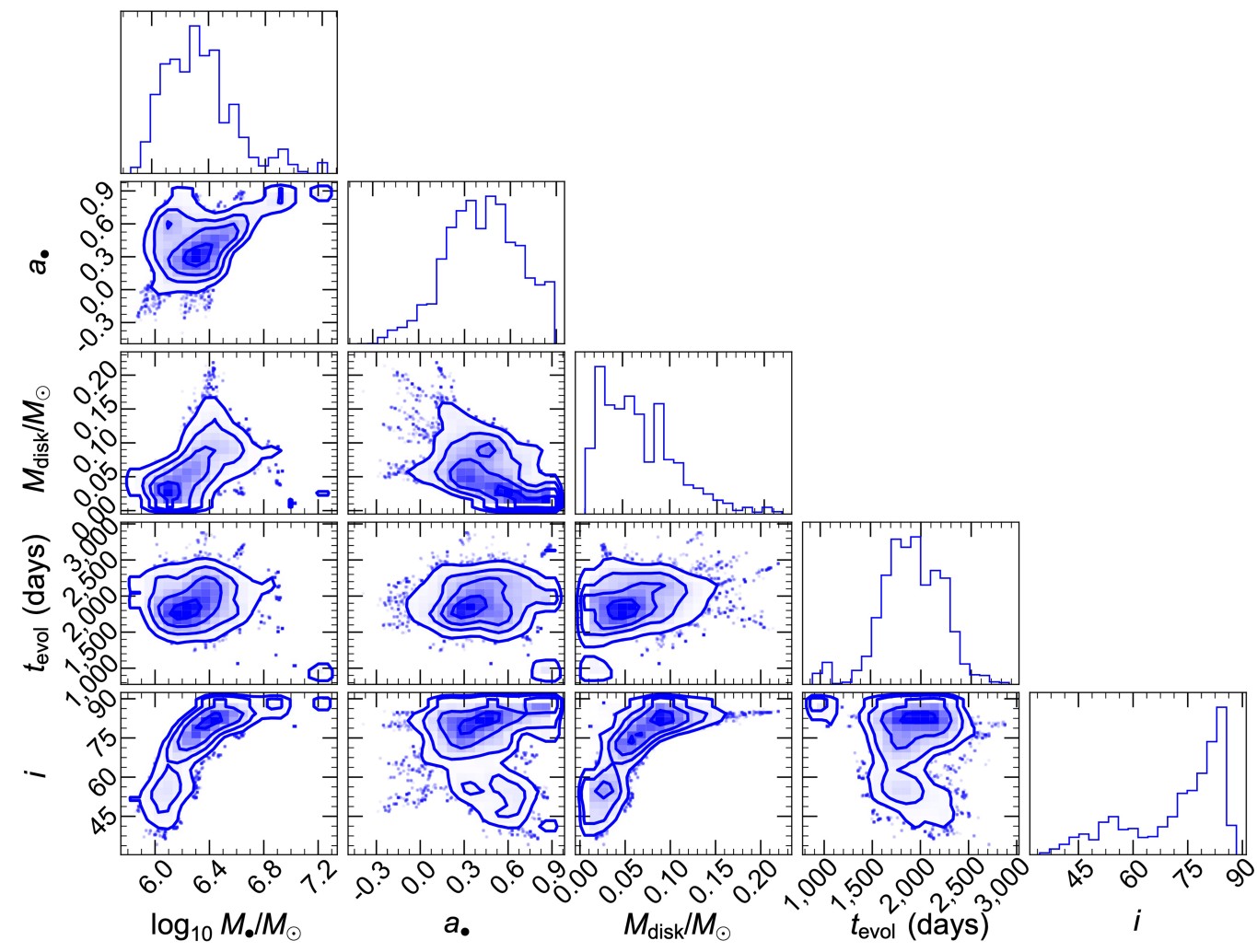

**Extended Data Fig. 8 | Parameter constraints from the disk model.**
The posterior distributions of the model fit to AT2019qiz. The SMBH mass posterior ($M_\odot$) is consistent with all other observational constraints and all other parameter values are in the expected range for TDEs. The SMBH spin is denoted $a_\bullet$, $M_{\mathrm{disk}}$ is the initial disk mass, $t_{\mathrm{evol}}$ parameterizes the timescale of viscous spreading and $i$ is the inclination of the disk with respect to the observer.

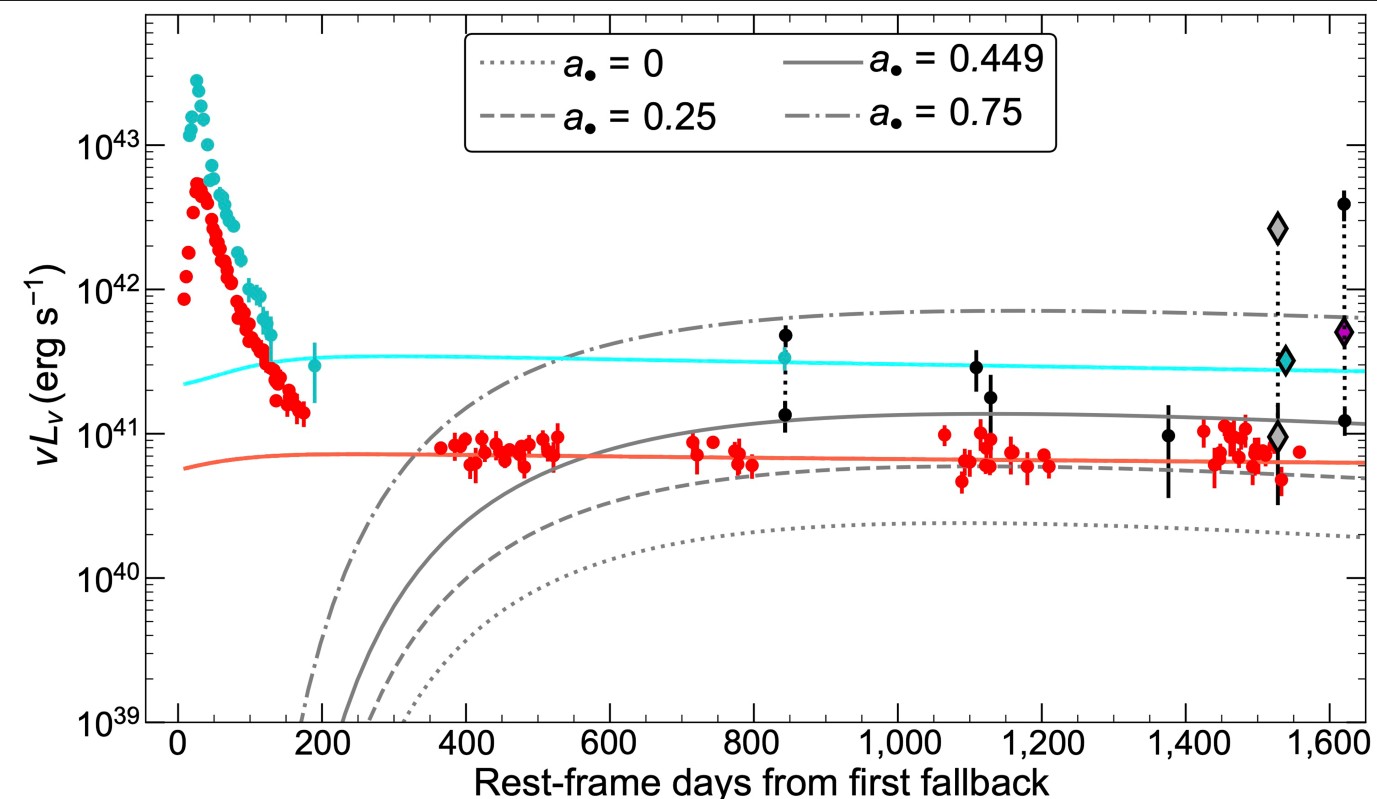

**Extended Data Fig. 9 | Examples of disk model light curves for four different SMBH spin values.** All other parameters are fixed to the posterior medians. The colours are the same as in Fig. 4. In optical and UV bands, varying the spin produces imperceptible changes in the light curves, but in the X-ray band, the changes are pronounced. Physically, this is a result of the exponential sensitivity of the X-ray flux on the inner disk temperature, whereas the optical and UV luminosity is sensitive only to the disk structure at larger radii.

**Extended Data Table 1 | Estimates of the SMBH mass in AT2019qiz**

| Method | $\log(M_\bullet/M_\odot)$ | Uncertainty | Source |
|---|---|---|---|
| Galaxy scaling relations | | | |
| $M_\bullet - \sigma$ [94] | 6.54 | 0.32 | [16] |
| $M_\bullet - \sigma$ [95] | 6.24 | 0.48 | [16] |
| $M_\bullet - \sigma$ [96] | 5.82 | 0.41 | [16] |
| Light curve modeling | | | |
| MOSFIT [97,98] | 5.89 | 0.21 | [16] |
| MOSFIT [97,98] | 6.14 | 0.1 | [17] |
| MOSFIT [97,98] | 6.22 | 0.2 | [99] |
| TDEMASS [100] | 6.18 | 0.07 | [100] |
| TDE scaling relations | | | |
| $L_{\mathrm{plateau}}$ [19] | 5.42 | +0.58, -0.45 | [19] |
| $L_{\mathrm{peak}}$ [19] | 6.13 | 0.5 (sys. only) | [19] |
| $E_{\mathrm{rad}}$ [19] | 6.35 | 0.5 (sys. only) | [19] |
| **Disk modeling** [25] | **6.3** | **+0.3, −0.2** | **This work** |

Errors represent the 1σ uncertainty including statistical and systematic errors unless specified otherwise.