## [Peer Review File · Nature]

Manuscript Title: Quasi-periodic X-ray eruptions years after a nearby tidal disruption event

Reviewer Comments & Author Rebuttals

Reviewer Reports on the Initial Version:

Referees' comments:

Referee #1 (Remarks to the Author):

Quasi-periodic X-ray eruptions years after a nearby tidal disruption event

This paper describes quasi-periodic eruptions (QPEs) newly detected in a galaxy that was previously seen to have undergone an optical tidal disruption event (TDE). This provides the clearest relation yet between QPEs and TDEs, which is important for calculating the rates of EMRIs that may be detected in future gravitational wave detectors such as LISA. It also provides further support for the interpretation that QPEs are produced by the repeated interaction of an orbiting body with the accretion disc. The paper is well written and succinct and I have only minor comments and requests for clarifications.

p.7 - is the statement that the 'orbiting body must always cross the disc' also valid in the case of an elliptical disc as sometimes inferred for TDEs?

p.8 - could the authors give more quantitative details in methods of the relationship between T_{rec} and spin and the other parameters? In particular can they demonstrate that the precession timescale is less than $10 T_{\text{rec}}$ for $\text{spin} > 0.5$?

Figure 2: If the apparent black-body radius of the emitting region decreases in the final decay point can this be explained in terms of the EMRI model?

Fig 3.: The addition of the AT2019qiz point, at least by eye, makes the correlation between eruption duration and the recurrence time seem very compelling. Can the authors give a statistical measure for the significance of this correlation? Can the correlation be explained by the EMRI model?

Fig 4 and Ext Fig 9: Can the authors explain more clearly in methods how the host galaxy light was subtracted or modelled during the disc fits?

Methods 1.1.4 - Fig 6. should read "Extended figure 6"

Methods 2.3 What is meant by an orbiting "third" body? Do the authors mean that the object passing through the disc is in addition to the star which caused the original TDE? If so, could it not be the remnant of the same star that is causing the QPEs?

Ext Fig. 8: Please define t_{evol} ?

Referee #2 (Remarks to the Author):

The authors have found an exciting new association between two different transient phenomena in SMBH accretion astrophysics: TDEs and QPEs. As the authors note, there have been past hints of such an association, but the novel observations in this paper provide, for the first time, clear proof that some TDEs exhibit QPEs (or, alternatively, that some QPEs are found in association with TDEs). In my opinion, this novel discovery warrants publication in *Nature*, as it links two different transients in a way that greatly improves our knowledge of at least one (QPEs, whose origin has been hotly debated), and may have significant implications for our understanding of the other (TDEs). The methodology and statistical significance of the discovery (i.e. the TDE-QPE association) are convincing.

I have some concerns about one component of the modeling/interpretation, which I will discuss presently, as well as a longer list of questions/comments below, but provided these are addressed, I will happily endorse the publication of this work in *Nature*.

My only major concern with the interpretation given in the paper concerns the compatibility between (i) the theoretical models for a planar, viscously spreading disk that are used here, and (ii) the interpretation that variation in the recurrence time originates from global Lense-Thirring precession of the late-time TDE accretion disk. There are two problems here:

1. First, I am somewhat skeptical that global precession will still be happening at such late times in the disk evolution. Past works, e.g. Stone & Loeb 2012 or Franchini et al. 2016: <https://ui.adsabs.harvard.edu/abs/2016MNRAS.455.1946F/abstract> find that tilted TDE disks will typically align into the SMBH equatorial plane on timescales much shorter than 1000 days, shutting off precession, unless (A) the SMBH is slowly spinning and (B) angular momentum transport is quite weak in the disk, with a Shakura-Sunyaev alpha parameter well under 0.1 (see e.g. Franchini et al. 2016 Fig. 9). If both these conditions are met, then the alignment time can be postponed until after a few thousand days, as is required by observations. However, it is not at all clear if the precession timescale can remain short enough to explain inter-burst timing variations for a very low BH spin parameter. The authors need to better quantify the range of precession timescales they calculate and especially to determine the minimum spin value that is compatible with recurrence time variability.

2. Even if the disk can be globally precessing, this raises the question of how applicable the planar (implicitly equatorial) models of viscously spreading disks used in the interpretation are. Certainly predictions for X-ray luminosity will be dramatically different in a highly tilted disk, since the equatorial

ISCO will no longer be the relevant inner edge that dominates quasi-thermal X-ray production. I understand that the long-term viscous evolution of a globally tilted disk is non-trivial, and goes beyond the scope of a primarily observational paper, but if global precession remains a viable/preferred interpretation after checking the alignment timescales for minimum spin values (point 1 above), the paper needs to include substantially more caveats about the validity of aligned disk models for a tilted, precessing system.

Major comments/questions:

-P. 6 (and later): the early-time non-detection of QPEs with NICER is quite interesting. Using the spreading disk models, can this be used to constrain the time at which substantial material circularized? More specifically, if one takes an extreme case where material circularizes and feeds a spreading disk with a source term at $2 * R_{\text{tidal}}$, would the disk spread quickly enough to reach the putative orbit of the star/compact object that produces the late-time QPEs?

-P. 7: given the main comment I have above, I think all mention of fitted SMBH spin should be removed from the main text. The equatorial ISCO (used to measure this spin) has no relevance if the disk is highly inclined and precessing. It can still be discussed in the Methods, but needs to be caveated appropriately.

-Fig. 3: The doubled-up vertical axes (left hand side and right hand side) seem a little wasteful; I don't think it adds much to include both days and hours, and the figures would be better off bigger. Alternatively, one could add extra information. For example, if one assumes that the EMRI causing the QPEs is a stellar EMRI close to the main sequence mass-radius relationship, the BH mass plus a recurrence time picks a unique stellar density and therefore a unique EMRI mass. This could be an informative alternative for the right-hand-side y-axis on the right panel.

-P. 23: it is too strong to say that modest errors in f_{col} dominate the error budget for continuum fitting spin estimates. This may be true in some cases, but there are other circumstances (e.g. a tilted disk, as the authors invoke) where there may be larger systematic uncertainties.

-P. 23: please explicitly state the ansatz you are using for the stress/effective viscosity.

-P. 24: the authors need to give a quantitative range of disk global precession timescales, given the importance this has for explaining timing aperiodicities in their model.

-P. 24: if the authors wish to "rule out" classes of QPE models, they need to provide more detailed argumentation than what is given for the lensing / disk tearing scenarios. Please make these arguments stronger / more explicit or alternatively delete the claim.

Minor comments/questions:

-P. 3: a very early prediction of QPEs from star-disk crossing interactions was made by Dai & Blandford:

<https://ui.adsabs.harvard.edu/abs/2010MNRAS.402.1614D/abstract>
this early work should be cited.

-P. 4: in the abstract, the authors highlight the uniqueness of their discovery relative to past hints of QPEs in TDEs by using the phrase “spectroscopically confirmed TDE”. However, it seems to me that what really sets this discovery apart from past observations of e.g. GSN 069 (in terms of establishing a QPE-TDE connection) is not spectroscopy but rather photometry that captures the peak and even the rise of the original TDE flare. I recommend changing this phrase to a “TDE captured at peak light” or somesuch.

-P. 6: it is too strong to say that this result “explains” the common host galaxy populations of TDEs and QPEs, since the authors have not (yet) established that most QPEs originate from TDEs. Please soften the language here.

-P. 7: the fitted disk mass is the *initial* disk mass, right? If so, it needs to be specified. If not, it should be clear at what time this is the present-day disk mass.

-P. 7: another explanation for flaring when a stellar-mass body crosses an SMBH disk is shock breakout, e.g. Tagawa & Haiman 2023 (cited later in the paper). This should be mentioned here.

-P. 8: I do not think it is accurate to say that an $r^{-1/2}$ profile is “best-fitting” since it is not as if the authors have varied the power law index of $\Sigma(R)$. Rather they have assumed a power law index and modeled the results, fitting other parameters along the way. Unless I have misunderstood this point, the wording should be changed here.

-Fig. 4: please be explicit about the meaning of the vertical black lines in the bottom (Σ) panel. I also suggest showing the surface density profile at the time when the disk outer edge first reaches the $P = 48.4$ h radius, and labeling how many days into the disk evolution this is achieved (I think this may be a useful guide/motivation for future calculations and observations).

-P. 19: “a fact of” should be “a factor of”

-P. 22: I do not think much more needs to be done to establish that the QPE signals are not just random variability, but wouldn't a logical comparison here be between the power spectra of the QPE X-ray light curve and power spectra from a larger sample of e.g. AGN ?

-P. 23: the statement that disk temperature only probes the product $W_{\text{phi}}^{\text{r}} * \Sigma$ is only true for steady-state disks, please make this clear.

Author Rebuttals to Initial Comments:

Referee #1 (Remarks to the Author):

Quasi-periodic X-ray eruptions years after a nearby tidal disruption event

This paper describes quasi-periodic eruptions (QPEs) newly detected in a galaxy that was previously seen to have undergone an optical tidal disruption event (TDE).

This provides the clearest relation yet between QPEs and TDEs, which is important for calculating the rates of EMRIs that may be detected in future gravitational wave

detectors such as LISA. It also provides further support for the interpretation that QPEs are produced by the repeated interaction of an orbiting body with the accretion disc. The paper is well written and succinct and I have only minor comments and requests for clarifications.

We thank the referee for their constructive and positive comments. We have revised the paper following their suggestions and summarise our changes below

p.7 - is the statement that the 'orbiting body must always cross the disc' also valid in the case of an elliptical disc as sometimes inferred for TDEs?

We have rephrased language like “must” to softer statements like “is expected to”. In the Methods section discussing constraints on QPE models, we have added the (somewhat fine-tuned) requirements on disk/orbital geometry in order to avoid crossing: the disk ellipticity must be at least $>\sim 0.7$, and the orbit aligned along the semi-minor axis of the disk.

Simulations of TDE disks, evolved for more than a few debris orbital timescales, generally show an evolution towards circularity on timescales much shorter than the time when we observe QPEs (e.g. Bonnerot & Lu 2020, Curd 2021). Furthermore, given that the evolutionary timescale of disc material at a given radius scales as $R^{3/2}$ (on orbital grounds), we would expect the eccentricity of the flow to drop with time, as the inner material spreads quicker relative to the outer regions. These arguments suggest that a circular disk should be a very good approximation at late times.

p.8 - could the authors give more quantitative details in methods of the relationship

between T_{rec} and spin and the other parameters? In particular can they demonstrate that the precession timescale is less than $10 T_{\text{rec}}$ for $\text{spin} > 0.5$?

We have substantially revised the aspects of the paper dealing with precession, following suggestions from both referees. We have added a new section to the Methods, “Precession timescales” that lays out the range of precession timescales for different spins and density profiles, and the alignment timescales of the disk with the SMBH spin. At the high spin needed to explain the short-timescale timing variations, the disk would align too quickly, making it difficult to confidently attribute timing variations to precession.

We have also replaced the paragraph on precession in the main text with a new paragraph that is much more cautious about whether precession can explain the observed timing residuals, noting that the disk will likely align on shorter timescales. We now also note the new results from Yao et al (<https://ui.adsabs.harvard.edu/abs/2024arXiv240714578Y/abstract>) which show that dynamics of the shocked stellar debris can also introduce timing variations.

Figure 2: If the apparent black-body radius of the emitting region decreases in the final decay point can this be explained in terms of the EMRI model?

The EMRI model from Linial & Metzger produces radiation in shocked, expanding ejecta produced in each collision. As the ejecta expand, we expect they would ultimately become optically thin and the photosphere would recede. This is analogous to observations of supernovae. We have clarified this point in the caption and in the main text.

Fig 3.: The addition of the AT2019qiz point, at least by eye, makes the correlation

between eruption duration and the recurrence time seem very compelling. Can the authors give a statistical measure for the significance of this correlation? Can the correlation be explained by the EMRI model?

We have performed a correlation analysis on the QPE data including AT2019qiz, and find strong Bayesian evidence for a positive correlation with a slope consistent with the literature. We have added a short subsection to the Methods.

The EMRI model does predict a correlation between duration and recurrence time, with additional scatter due to the BH mass and the radius of the star (Linial & Metzger). However, we are cautious to over-interpret this here.

Fig 4 and Ext Fig 9: Can the authors explain more clearly in methods how the host galaxy light was subtracted or modelled during the disc fits?

We have added text to the methods section detailing the approach to dealing with host galaxy light for the optical, NUV and FUV data in the relevant Methods sections. Optical and NUV photometry have been host-subtracted, while we expect that the host contribution in the FUV is not significant.

Methods 1.1.4 - Fig 6. should read "Extended figure 6"

This has been corrected.

Methods 2.3 What is meant by an orbiting "third" body? Do the authors mean that the object passing through the disc is in addition to the star which caused the original TDE? If so, could it not be the remnant of the same star that is causing the QPEs?

Indeed we meant another object in addition to the star that was disrupted in 2019. If the disrupted star approached the SMBH on a typical parabolic orbit and was partially disrupted, the remnant would end up on a very eccentric orbit with a period much longer than the QPE timescale (possibly > 1000 yrs, Cufari et al 2022). If we instead consider the star causing the QPEs, it could not have been involved in the TDE because the orbital radius with the QPE

period in AT2019qiz is about an order of magnitude larger than the tidal radius. A star on a mildly eccentric orbit that does cross the tidal radius and get repeatedly disrupted is one proposed explanation for QPEs (Lu & Quataert 2023), but in this case a small amount of matter is stripped with each pericenter passage, and it cannot generate a single large flare resembling a ‘classic’ TDE. In summary, it is very difficult to construct an orbit that produces both a large-amplitude TDE and short-period QPEs. There is a discussion of this in section 1 of Linial & Metzger 2023, and Figure 7 of Lu & Quataert gives a nice illustration of the orbits. We have added a note of this to Methods 2.3.

Ext Fig. 8: Please define t_{evol} ?

We apologise for this original omission, this has now been defined explicitly in terms of the turbulent stress amplitude in the extended information. In addition, we now define all parameters in the figure caption.

Referee #2 (Remarks to the Author):

The authors have found an exciting new association between two different transient phenomena in SMBH accretion astrophysics:

TDEs and QPEs. As the authors note, there have been past hints of such an association, but the novel observations in this paper provide, for the first time, clear proof that some TDEs exhibit QPEs (or, alternatively, that some QPEs are found in association with TDEs). In my opinion, this novel discovery warrants publication in Nature, as it links two different transients in a way that greatly improves our knowledge of at least one (QPEs, whose origin has been hotly debated), and may have significant implications for our understanding of the other (TDEs). The methodology and statistical significance of the discovery (i.e. the TDE-QPE association) are convincing.

I have some concerns about one component of the modeling/interpretation, which I will discuss presently, as well as a longer list of questions/comments below, but provided these are addressed, I will happily endorse the publication of this work in Nature.

We thank the referee for their constructive and positive comments. We have revised the paper following their suggestions and summarise our changes below

My only major concern with the interpretation given in the paper concerns the compatibility between (i) the theoretical models for a planar, viscously spreading disk that are used here, and (ii) the interpretation that variation in the recurrence time originates from

global Lense-Thirring precession of the late-time TDE accretion disk. There are two problems here:

1. First, I am somewhat skeptical that global precession will still be happening at such late times in the disk evolution. Past works, e.g. Stone & Loeb 2012 or Franchini et al. 2016:

<https://ui.adsabs.harvard.edu/abs/2016MNRAS.455.1946F/abstract>

find that tilted TDE disks will typically align into the SMBH equatorial plane on timescales much shorter than 1000 days, shutting off precession, unless (A) the SMBH is slowly spinning and (B) angular momentum transport is quite weak in the disk, with a Shakura-Sunyaev alpha parameter well under 0.1 (see e.g. Franchini et al. 2016 Fig. 9). If both these conditions are met, then the alignment time can be postponed until after a few thousand days, as is required by observations. However, it is not at all clear if the precession timescale can remain short enough to explain inter-burst timing variations for a very low BH spin parameter. The authors need to better quantify the range of precession timescales they calculate and especially to determine the minimum spin value that is compatible with recurrence time variability.

2. Even if the disk can be globally precessing, this raises the question of how applicable the planar (implicitly equatorial) models of viscously spreading disks used in the interpretation are. Certainly predictions for X-ray luminosity will be dramatically different in a highly tilted disk, since the equatorial ISCO will no longer be the relevant inner edge that dominates quasi-thermal X-ray production. I understand that the long-term viscous evolution

of a globally tilted disk is non-trivial, and goes beyond the scope of a primarily observational paper, but if global precession remains a viable/preferred interpretation after checking the alignment timescales for minimum spin values (point 1 above), the paper needs to include substantially more caveats about the validity of aligned disk models for a tilted, precessing system.

We have substantially revised the aspects of the paper dealing with precession. We have added a new section to the Methods, “Precession timescales” that lays out the range of precession timescales for different spins and density profiles, and now discusses the alignment timescales too. We use the helpful reference suggested by the referee to check the alignment time. As they predict, at the high spin needed to explain the short-timescale timing variations, the disk would align too quickly. We state this explicitly alongside the other caveat noted in point 2.

We have also replaced the paragraph on precession in the main text with a new paragraph that is much more cautious about whether precession can explain the observed timing residuals, noting that the disk will likely align on shorter timescales. We now also note the new results from Yao et al (<https://ui.adsabs.harvard.edu/abs/2024arXiv240714578Y/abstract>) which show that dynamics of the shocked stellar debris can also introduce timing variations.

Major comments/questions:

-P. 6 (and later): the early-time non-detection of QPEs with NICER is quite interesting. Using the spreading disk models, can this be used to constrain the time at which substantial material circularized? More specifically, if one takes an extreme case where material circularizes and feeds a spreading disk with a source term at $2 * R_{\text{tidal}}$, would the disk spread quickly enough to reach the putative orbit of the star/compact object that produces the late-time QPEs?

We have added text to the Methods section (on NICER) noting that these observations are unfortunately not very constraining for probing the properties of the spreading disk at early times, and therefore the onset of QPEs. These data span the peak of the optical light curve, during which time this TDE had an extended reprocessing atmosphere and appears optically thick to X-rays.

As there is no early time data with which to constrain the starting radius of the disk ring, and we do not wish to over-constrain the model in a time period with no data, we leave this radius free and do not prescribe it to a given circularisation scale. We can confirm that all disk models which are too small to reproduce the QPEs at early times all have sufficiently short evolutionary timescales to reach the EMRI-interception scale by 800 days, when the optical/UV plateau is present and the disk structure can be constrained by observations.

-P. 7: given the main comment I have above, I think all mention of fitted SMBH spin should be removed from the main text. The equatorial ISCO (used to measure this spin) has no relevance if the disk is highly inclined and precessing. It can still be discussed in the Methods, but needs to be caveated appropriately.

Constraints on the SMBH spin have been removed from the main text. We have also added this caveat both to the section on Disk Modelling and the new section “Precession timescales”.

-Fig. 3: The doubled-up vertical axes (left hand side and right hand side) seem a little wasteful; I don't think it adds much to include both days and hours, and the figures would be better off bigger. Alternatively, one could add extra information. For example, if one assumes that the EMRI causing the QPEs is a stellar EMRI close to the main sequence mass-radius relationship, the BH mass plus a recurrence time picks a unique stellar density and therefore a unique EMRI mass. This could be an informative alternative for the right-hand-side y-axis on the right panel.

We have removed the doubled axes on these plots. Increasing the size of the plotting area has improved the figure.

-P. 23: it is too strong to say that modest errors in f_{col} dominate the error budget for continuum fitting spin estimates. This may be

true in some cases, but there are other circumstances (e.g. a tilted disk, as the authors invoke) where there may be larger systematic uncertainties.

We have changed the wording here to say instead that modest uncertainties in f_{col} can lead to “substantial” uncertainties in spin.

-P. 23: please explicitly state the ansatz you are using for the stress/effective viscosity.

We thank the reviewer for pointing out this omission. This definition has been added to the supplementary material.

-P. 24: the authors need to give a quantitative range of disk global precession timescales, given the importance this has for explaining timing aperiodicities in their model.

As noted above, we have now stated explicitly the ranges of timescales for different spins, as well as the inconsistency that arises if the disk aligns on timescales shorter than the current age of the TDE. We have toned down the discussion of precession in the main text.

-P. 24: if the authors wish to “rule out” classes of QPE models, they need to provide more detailed argumentation than what is given for the lensing / disk tearing scenarios. Please make these arguments stronger / more explicit or alternatively delete the claim.

We have deleted the sentence on ruling out these models, and now simply note the observables in AT2019qiz that differ from the expectations in these scenarios without making such strong claims.

Minor comments/questions:

-P. 3: a very early prediction of QPEs from star-disk crossing interactions was made by Dai & Blandford:

<https://ui.adsabs.harvard.edu/abs/2010MNRAS.402.1614D/abstract>

this early work should be cited.

We were not aware of this earlier work, and thank the referee for highlighting it. We have added a citation.

-P. 4: in the abstract, the authors highlight the uniqueness of their discovery relative to past hints of QPEs in TDEs by using the phrase “spectroscopically confirmed TDE”. However, it seems to me that what really sets this discovery apart from past observations of e.g. GSN 069 (in terms of establishing a QPE-TDE connection) is not spectroscopy but rather photometry that captures the peak and even the rise of the original TDE flare. I recommend changing this phrase to a “TDE captured at peak light” or somesuch.

We have changed this to “a spectroscopically confirmed TDE or an optical TDE captured at peak brightness”, as we think the spectroscopic classification is also important to fully exclude any other kind of optical variability that could mimic a TDE (e.g. coincident nuclear supernova).

-P. 6: it is too strong to say that this result “explains” the common host galaxy populations of TDEs and QPEs, since the authors have not (yet) established that most QPEs originate from TDEs. Please soften the language here.

We have removed this statement.

-P. 7: the fitted disk mass is the *initial* disk mass, right? If so, it needs to be specified. If not, it should be clear at what time this is the present-day disk mass.

This does refer to the initial disk mass. This was mentioned only in the methods but we have now specified in the main text too.

-P. 7: another explanation for flaring when a stellar-mass body crosses an SMBH disk is shock breakout, e.g. Tagawa & Haiman 2023 (cited later in the paper). This should be mentioned here.

We have added this to the main text here.

-P. 8: I do not think it is accurate to say that an $r^{-1/2}$ profile is “best-fitting” since it is not as if the authors have varied the power law index of $\Sigma(R)$. Rather they have assumed a power law index and modeled the results, fitting other parameters along the way. Unless I have misunderstood this point, the wording should be changed here.

The reviewer is correct, and this original phrasing was inaccurate. We have amended this discussion in the new version of the manuscript.

-Fig. 4: please be explicit about the meaning of the vertical black lines in the bottom (Sigma) panel. I also suggest showing the surface density profile at the time when the disk outer edge first reaches the $P = 48.4$ h radius, and labeling how many days into the disk evolution this is achieved (I think this may be a useful guide/motivation for future calculations and observations).

We have added explanations of these lines to the figure caption.

The time at which the bulk disk density profile reaches a given radius is poorly constrained at times earlier than the onset of the optical/UV plateau phase and the detection of X-rays (i.e., ~500-800 days). While at earlier times (~100 days) the bulk of the disk density posteriors are within the interception radius, we cannot be sure this is robust owing to the lack of observational constraints. As this transitional time is more sensitive to the (artificial) initial condition than the later behaviour of the disk, we think it is safer not to include this analysis in the manuscript.

-P. 19: “a fact of” should be “a factor of”

Corrected.

-P. 22: I do not think much more needs to be done to establish that the QPE signals are not just random variability, but wouldn't a logical comparison here be between the power spectra of the QPE X-ray light curve and power spectra from a larger sample of e.g. AGN ?

This is a good suggestion, however the number of consecutive QPEs observed so far is quite low to construct a power spectrum. We are obtaining more data that will enable this in the future. For now we agree with the referee that the QPE signals are clearly not random variability!

-P. 23: the statement that disk temperature only probes the product $W_{\text{phi}}^r * \Sigma$ is only true for steady-state disks, please make this clear.

We have double-checked that this statement is also true when the disk is not in a steady state. This is discussed in detail in Balbus & Papaloizou (1999; section 3.1 leading up to equation 46 and subsequent text). While the above reference uses a Newtonian formalism of gravity, it can be shown that the same result holds in General Relativity.

Reviewer Reports on the First Revision:

Referees' comments:

Referee #2 (Remarks to the Author):

I thank the authors for addressing my concerns (and those of the other referee). In my opinion, this paper is now appropriate for publication in Nature.